**Analysis**

# Comparison of transformations for single-cell RNA-seq data

Constantin Ahlmann-Eltze ●[1,2] ✉ & Wolfgang Huber ●[1]

The count table, a numeric matrix of genes × cells, is the basic input data structure in the analysis of single-cell RNA-sequencing data. A common preprocessing step is to adjust the counts for variable sampling efficiency and to transform them so that the variance is similar across the dynamic range. These steps are intended to make subsequent application of generic statistical methods more palatable. Here, we describe four transformation approaches based on the delta method, model residuals, inferred latent expression state and factor analysis. We compare their strengths and weaknesses and find that the latter three have appealing theoretical properties; however, in benchmarks using simulated and real-world data, it turns out that a rather simple approach, namely, the logarithm with a pseudo-count followed by principal-component analysis, performs as well or better than the more sophisticated alternatives. This result highlights limitations of current theoretical analysis as assessed by bottom-line performance benchmarks.

Single-cell RNA-sequencing (RNA-seq) count tables are heteroskedastic. In particular, counts for highly expressed genes vary more than for lowly expressed genes. Accordingly, a change in a gene's counts from 0 to 100 between different cells is more relevant than, say, a change from 1,000 to 1,100. Analyzing heteroskedastic data is challenging because standard statistical methods typically perform best for data with uniform variance.

One approach to handle such heteroskedasticity is to explicitly model the sampling distributions. For data derived from unique molecular identifiers (UMIs), a theoretically and empirically well-supported model is the gamma-Poisson distribution (also referred to as the negative binomial distribution)[1–3], but parameter inference can be fiddly and computationally expensive[4,5]. An alternative choice is to use variance-stabilizing transformations as a preprocessing step and subsequently use the many existing statistical methods that implicitly or explicitly assume uniform variance for best performance[3,6].

Variance-stabilizing transformations based on the delta method[7] promise an easy fix for heteroskedasticity if the variance predominantly depends on the mean. Instead of working with the raw counts $Y$, we apply a non-linear function $g(Y)$ designed to make the variances (and possibly, higher moments) more similar across the dynamic range of

the data[8]. The gamma-Poisson distribution with mean $\mu$ and overdispersion $\alpha$ implies a quadratic mean–variance relationship $\mathbb{V}\mathrm{ar}[Y] = \mu + \alpha\mu^2$. Here, the Poisson distribution is the special case where $\alpha = 0$ and $\alpha$ can be considered a measure of additional variation on top of the Poisson. Given such a mean–variance relationship, applying the delta method produces the variance-stabilizing transformation

$$g(y) = \frac{1}{\sqrt{\alpha}}\operatorname{acosh}(2\alpha y + 1).\qquad(1)$$

Supplementary Information A1 shows the derivation. Practitioners often use a more familiar functional form, the shifted logarithm

$$g(y) = \log(y + y_0).\qquad(2)$$

This approximates equation (1), in particular if the pseudo-count is $y_0 = 1/(4\alpha)$ (Supplementary Information A2).

An additional requirement is posed by experimental variations in sampling efficiency and different cell sizes[9], which manifest themselves in varying total numbers of UMIs per cell. Commonly, a so-called size factor $s$ is determined for each cell and the counts are divided by it

[1]Genome Biology Unit, EMBL, Heidelberg, Germany. [2]Faculty of Biosciences, Heidelberg University, Heidelberg, Germany. ✉e-mail: constantin.ahlmann@embl.de

before applying the variance-stabilizing transformation: for example, $\log(y/s + y_0)$[6,10,11]. There is a variety of approaches to estimate size factors from the data. Conventionally, they are scaled to be close to 1 (for example, by dividing them by their mean), such that the range of the adjusted counts is about the same as that of the raw counts. The simplest estimate of the size factor for cell $c$ is

$$s_c = \frac{\sum_g y_{gc}}{L}, \tag{3}$$

where the numerator is the total number of UMIs for cell $c$, $g$ indexes the genes and $L = (\text{no. cells})^{-1} \sum_{gc} y_{gc}$ is the average across all cells of these numerators.

Sometimes, a fixed value is used instead for $L$. For instance, Seurat uses $L = 10{,}000$, others[12] have used $L = 10^6$ calling the resulting values $y_{gc}/s_c$ counts per million (CPM). Even though the choice of $L$ may seem arbitrary, it matters greatly. For example, for typical droplet-based single-cell data with sequencing depth of $\sum_g y_{gc} \approx 5{,}000$, using $L = 10^6$ and then transforming to $\log(y_{gc}/s_c + 1)$ is equivalent to setting the pseudo-count to $y_0 = 0.005$ in equation (2). This amounts to assuming an overdispersion of $\alpha = 50$, based on the relation between pseudo-count and overdispersion explained in Supplementary Information A2. That is two orders of magnitude larger than the overdispersions seen in typical single-cell datasets. In contrast, using the same calculation, Seurat's $L = 10{,}000$ implies a pseudo-count of $y_0 = 0.5$ and an overdispersion of $\alpha = 0.5$, which is closer to overdispersions observed in real data. Yet, choosing $L$ or $y_0$ is unintuitive. Instead, we recommend parameterizing the shifted logarithm transformation in terms of the typical overdispersion, using the relation $y_0 = 1/(4\alpha)$ motivated above.

Hafemeister and Satija[13] suggested a different approach to variance stabilization based on Pearson residuals

$$r_{gc} = \frac{y_{gc} - \hat{\mu}_{gc}}{\sqrt{\hat{\mu}_{gc} + \hat{\alpha}_g \hat{\mu}_{gc}^2}}, \tag{4}$$

where $\hat{\mu}_{gc}$ and $\hat{\alpha}_g$ come from fitting a gamma-Poisson generalized linear model (GLM),

$$Y_{gc} \sim \text{gamma-Poisson}\,(\mu_{gc}, \alpha_g)$$
$$\log(\mu_{gc}) = \beta_{g,\text{intercept}} + \beta_{g,\text{slope}} \log(s_c). \tag{5}$$

Here, $s_c$ is again the size factor for cell $c$, and $\beta_{g,\text{intercept}}$ and $\beta_{g,\text{slope}}$ are intercept and slope parameters for gene $g$. Note that the denominator in equation (4) is the s.d. of a gamma-Poisson random variable with parameters $\hat{\mu}_{gc}$ and $\hat{\alpha}_g$.

A third set of transformations infers the parameters of a postulated generative model, aiming to estimate so-called latent gene expression values based on the observed counts. A prominent instance of this approach is Sanity, a fully Bayesian model for gene expression[14]. It infers latent gene expression using a method that resembles a variational mean-field approximation for a log-normal Poisson mixture model. Sanity comes in two flavors: Sanity Distance calculates the mean and s.d. of the posterior distribution of the logarithmic gene expression; based on these, it calculates all cell-by-cell distances, from which it can find the $k$-nearest neighbors ($k$-NN) of each cell. Sanity MAP (maximum a posteriori) ignores the inferred uncertainty and returns the maximum of the posterior as the transformed value. A related tool is Dino, which fits mixtures of gamma-Poisson distributions and returns random samples from the posterior[15]. Normalisr is a tool primarily designed for frequentist hypothesis testing[16], but as it infers logarithmic latent gene expression, it might also serve as a generic preprocessing method. Normalisr returns the minimum mean square error estimate for each count assuming a binomial generative model.

In this work, we analyze transformations for preprocessing UMI-based single-cell RNA-seq data based on each of these approaches. We will first contrast the conceptual differences between them. In a second part, we benchmark the empirical performance of all approaches and provide guidelines for practitioners to choose among the methods. In the benchmarks, we also include a fourth preprocessing approach that is not transformation-based and directly produces a low-dimensional latent space representation of the cells: factor analysis for count data based on the (gamma-)Poisson sampling distribution. An early instance of this approach, called GLM PCA, was presented by Townes[4] and applied to biological data by Townes et al.[17]. Recently, Agostinis et al.[18] presented an optimized implementation called NewWave.

## Results

There are multiple formats for each of the four approaches:

- Among the delta method-based variance-stabilizing transformations, we considered the acosh transformation equation (1), the shifted logarithm equation (2) with pseudo-count $y_0 = 1$ or $y_0 = 1/(4\alpha)$ and the shifted logarithm with CPM. In addition, we tested the shifted log transformation with highly variable gene selection (HVG), $z$ scoring (Z) and rescaling the output as suggested by Booeshaghi et al.[19].
- Among the residuals-based variance-stabilizing transformations, we considered the clipped and unclipped Pearson residuals (implemented by sctransform and transformGamPoi) and randomized quantile residuals. In addition, we tested the clipped Pearson residuals with HVG selection, $z$ scoring and an analytical approximation to the Pearson residuals suggested by Lause et al.[20].
- Among the latent gene expression-based transformations (Lat Expr), we considered Sanity Distance and Sanity MAP, Dino and Normalisr.
- Among the count-based factor analysis models (Count), we considered GLM PCA and NewWave.

Last, we include two methods as negative (Neg) controls in our benchmarks, for which we expect poor performance: the raw untransformed counts ($y$) and the raw counts scaled by the size factor ($y/s$).

### Conceptual differences

A known problem for variance-stabilizing transformations based on the delta method derives from the size factors. Figure 1a shows the first two principal components of a homogeneous solution of droplets encapsulating aliquots from the same RNA[21] for representative instances of the delta method-, residuals- and latent expression-based transformation approaches. Extended Data Fig. 1 shows the results for all transformations. Despite the size factor scaling, after the delta method-based transformation, the size factor remained a strong variance component in the data (Extended Data Fig. 1b). In contrast, the other transformations better mixed droplets with different size factors. Intuitively, the trouble for the delta method-based transformation stems from the fact that the division of the raw counts by the size factors scales large counts from droplets with large size factors and small counts from droplets with small size factors to the same value. This violates the assumption of a common mean–variance relationship. In Supplementary Information A3, we dissect this phenomenon more formally.

One of the motivations stated by Hafemeister and Satija[13] for the Pearson residuals-based variance-stabilizing transformation is that the delta method-based transformations fail to stabilize the variance of lowly expressed genes. Warton[22] provided a theoretical explanation for this fact. Indeed, Fig. 1b shows that the variance after transformation with a delta method-based variance-stabilizing transformation was practically zero for genes with a mean expression of <0.1. In contrast, after residuals-based transformation, the variance showed a weaker

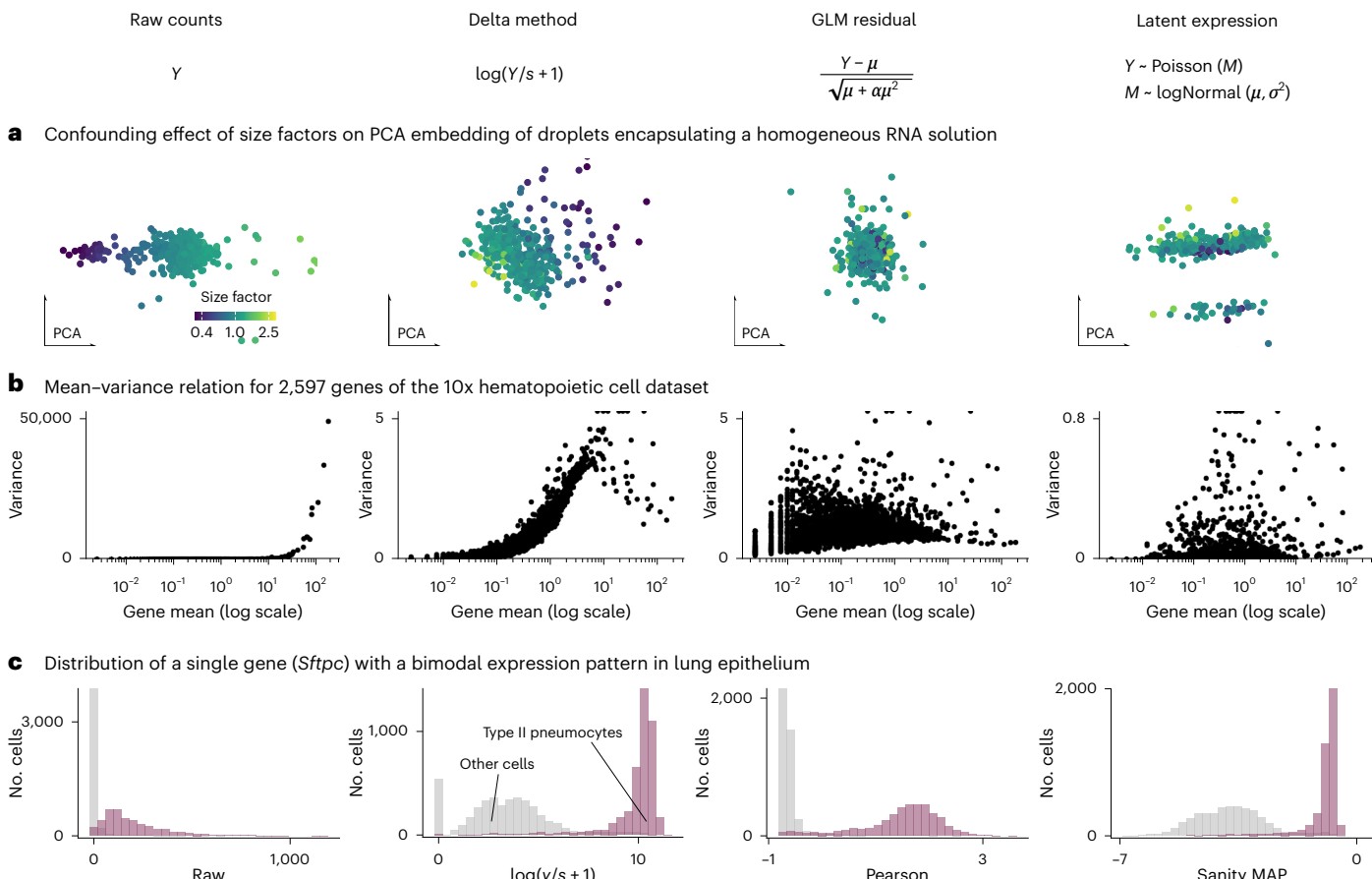

**Fig. 1 | Conceptual differences between variance-stabilizing transformations.** The four columns of this figure correspond to raw counts and transformation by shifted logarithm, clipped Pearson residuals and Sanity MAP. **a**, Scatter-plots of the first two principal components of data from droplets encapsulating a homogeneous RNA solution. Each point corresponds to a droplet and is colored by its size factor. **b**, Scatter-plots of the mean–variance relationship, where each point is a gene from a human hematopoietic cell dataset. Note that the *y* axis range differs between transformations and outliers are plotted on the edge of the plot. **c**, Histogram of the transformed values for Sftpc, a marker for type II pneumocytes that has a bimodal gene expression in mouse lung epithelium. Source data are provided.

dependence on mean expression, except for very lowly expressed genes whose variance is limited by the clipping step (compare Pearson and Pearson (no clip) in Extended Data Fig. 2). The results of the latent expression-based transformations were diverse, reflecting that these methods are not directly concerned with stabilizing the variance. Individual patterns ranged from higher variance for lowly expressed genes (Sanity Distance and Normalisr) to the opposite trend for Dino (Extended Data Fig. 2).

A peculiarity of the Pearson residuals is their behavior if a gene's expression strongly differs between cell subpopulations. Figure 1c shows a bimodal expression pattern of Sftpc, a marker for type II pneumocytes. Unlike the transformations based on the delta method or latent expression models, the Pearson residuals are an affine-linear transformation per gene (equation (4)) and thus cannot shrink the variance of the high-expression subpopulation more than that of the low-expression subpopulation (compare the Pearson residuals with $y/s$ in Extended Data Fig. 3). This can affect visualizations of such genes and, in principle, other analysis tasks such as detection of marker genes or clustering and classification of cells.

An alternative is to combine the idea of delta method-based variance-stabilizing transformations with the generalized linear model residuals approach by using non-linear residuals. We considered randomized quantile residuals[23] (Extended Data Fig. 4 shows how they are constructed). Like Pearson residuals, randomized quantile residuals stabilized the variance for small counts (Extended Data Fig. 2), but

in addition, they also stabilized the within-group variance if a gene's expression strongly differed across cells (Extended Data Fig. 3).

Such conceptual differences of the transformation approaches are important to understand when applying them to new data types or when developing new transformations; but for most practitioners, empirical performance will be of primary interest. We look at this in the next section.

## Benchmarks

There is no context-free measure of success for a preprocessing method, as it is contingent on the objectives of the subsequent analysis. For instance, if interest lies in identification of cell type-specific marker genes, one could assess the shape of distributions, such as in Fig. 1c, or the performance of a supervised classification method. Here, we considered the objective that arguably has been the main driver of single-cell RNA-seq development and applications so far: understanding the variety of cell types and states in terms of a lower-dimensional mathematical structure, such as a planar embedding, a clustering, trajectories, branches or combinations thereof. For all of these, one can consider the *k*-nearest neighbor (*k*-NN) graph as a fundamental data structure that encodes essential information. The next challenge is then the definition of 'ground truth'. We designed our benchmarks upon reviewing previous benchmarking approaches. For instance, Breda et al.[14] and Lause et al.[20] employed synthetic or semi-synthetic data. This is operationally attractive, but it is difficult to be certain

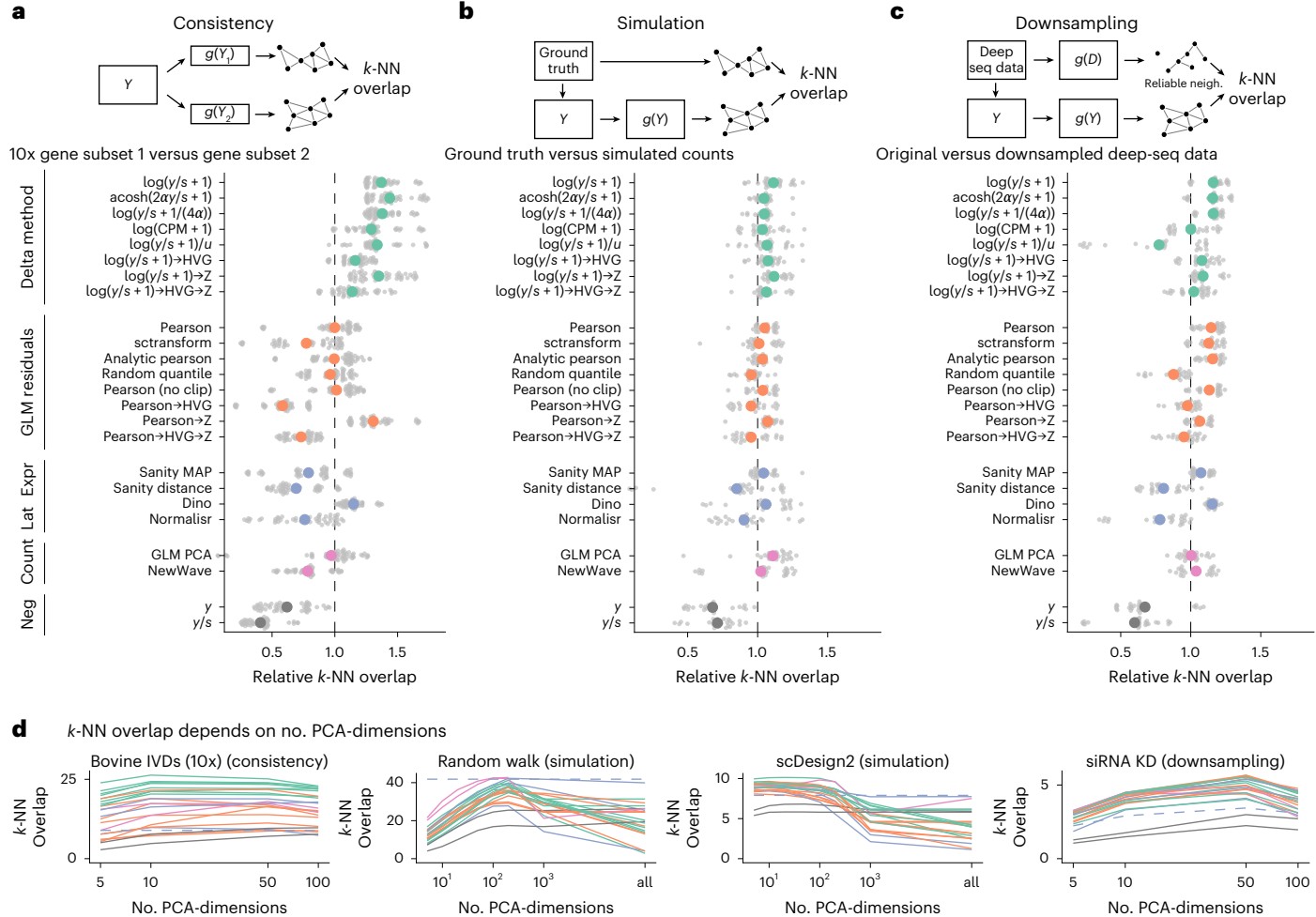

**Fig. 2 | Benchmark results. a**, Overlap between the $k$-NN inferred separately on two halves of the data. The colored points show the averages across ten datasets, each with five replicate random data splits (small, gray points). **b**, Overlap between $k$-NN inferred from simulated data and ground truth, using five simulation frameworks and five replicates per framework. **c**, Overlap between a reference $k$-NN graph (inferred using all transformations on deeply sequenced data and taking the intersection) and the $k$-NN inferred on data downsampled to match typical 10x data (5,000 counts per cell) for five datasets

with five replicates each. To compare and aggregate results across the different datasets, Relative overlap (**a**–**c**), which was computed by dividing, for each dataset, the overlap by its average across all transformations, fixing $k = 50$ and using a dataset-specific number of PCA dimensions (Extended Data Fig. 7 shows the underlying, unaggregated data). **d**, Overlap ($y$ axis) as a function of PCA dimensions ($x$ axis); the different transformation types are indicated by the colors, using the same palette as in **a**–**c**. The performance of Sanity Distance is shown as a dashed line. Source data are provided.

about biological relevance. Hafemeister and Satija[13] and Lause et al.[20] used qualitative inspection of non-linear dimension reduction plots. This can be informative, but is difficult to scale up and make objective. Germain et al.[24] compared how well the transformations recovered a priori assigned populations, defined either through FACS or by mixing different cell lines. This is conceptually clean, but restricts analysis to a limited range of datasets that also may only offer a caricature view of cell diversity.

For all our benchmarks, we applied the transformations to the raw counts of each dataset listed below, computed a lower-dimensional representation of the cells using principal-component analysis (PCA), identified the $k$-NNs of each cell as measured by Euclidean distance and, finally, computed the overlap of the thus obtained $k$-NN graph with a reference $k$-NN graph (Methods). We performed these three benchmarks:

- **Consistency.** We downloaded ten 10x datasets from the Gene Expression Omnibus (GEO) database. As there was no formal ground truth, we measured the consistency of the results (a necessary,

although not sufficient, condition for their goodness) by splitting the genes of each dataset into two disjoint subsets.
- **Simulation.** We used four different previously published simulation frameworks and one adapted by us to generate a diverse collection of datasets for which we had full access to the true $k$-NN graph.
- **Downsampling.** We used five deeply sequenced datasets based on mcSCRB and Smart-seq3, which we downsampled to sequencing depths typical for the 10x technology. We postulated that a proxy for ground truth could be constructed from the $k$-NN graph inferred from the deeply sequenced data intersected across all transformations which we call reliable nearest neighbors. To our knowledge, this work presents the first instance of such an approach.

Extended Data Fig. 5 and the Supplementary Information give an overview of the datasets.

We tested 22 transformations—where applicable with an over-dispersion fixed to 0, 0.05 and a gene-specific estimate from the

data—across four to eight settings for the number of dimensions of the PCA and measured the overlap with $k$ = 10, 50 and 100 nearest neighbors. In total, we collected more than 61,000 data points. In addition to the results highlighted in the following, we provide an interactive website with all results for all tested parameter combinations.

Figure 2 shows the aggregated results for the three benchmarks for $k$ = 50. Similar results were obtained for $k$ = 10 and $k$ = 100, shown in Extended Data Fig. 6.

In the consistency benchmark, the delta method-based transformations performed better than the other transformations (Fig. 2a).

On the simulated data, the differences between the transformations looked less pronounced in Fig. 2b than for the other two benchmarks; however, this is a result of the aggregated view. For each particular simulation framework, large differences between the transformations appeared, but the results varied from simulation to simulation framework (Extended Data Fig. 7b) and averaged out in the aggregated view.

The results of the downsampling benchmark (Fig. 2c) agreed well with the trends observed in the simulation and the consistency benchmark. This benchmark is particularly informative because the data had realistic latent structures that were reliably detectable through the high sequencing depth. The downsampling produced data that resembles the more common 10x data in many characteristics: for example, UMIs per cell, proportion of zeros in the data and mean–variance relationship (Supplementary Table 1 and Supplementary Fig. 1). The main difference was that the suitable (high sequencing depth per cell) datasets we could access mostly consisted of only a few hundred cells, except for the 4,298-cell short-interfering RNA KD dataset (Extended Data Fig. 5).

The results in Fig. 2 are on a relative scale, which hides the magnitude of the differences. In Extended Data Fig. 7, we show the underlying results for each dataset on an absolute scale. The range of $k$-NN overlaps was dataset dependent, ranging from 34 of 50 for the best performing transformation versus 9 of 50 for the negative control for the SUM149PT cell line consistency benchmark, to 2.9 of 50 versus 1.5 of 50 for the HEK downsampling benchmark. For the latter, the overall small overlaps were due to small sets of reliable nearest neighbors (Extended Data Fig. 8a,b). We also ran a version of the downsampling benchmark that only used the top two transformations per approach (Extended Data Fig. 8c,d), which increased the number of reliable nearest neighbors and confirmed the trends we saw in the full version.

In addition to the $k$-NN overlap with the ground truth, we also calculated the adjusted Rand index (ARI) and the adjusted mutual information (AMI) for the five simulation frameworks. Extended Data Fig. 9a,b shows the aggregated results, which were similar to the results for the $k$-NN overlap (Fig. 2b). Extended Data Fig. 9c,d show that the ARI and AMI had a larger dynamic range than the $k$-NN overlap for datasets with a small number of distinct clusters; however, for datasets with a complex latent structure, the $k$-NN overlap was more informative, which may reflect limitations of ARI and AMI to assess the recovery of gradual changes typical for many biological tissues.

The Random Walk simulation reproduced the benchmark based on which Breda et al.[14] argued that Sanity was the best method for identifying the $k$-NN of a cell (Fig. 5a of their paper). We found that the delta method-based and residuals-based variance-stabilizing transformations performed as well in this benchmark if we projected the cells to a lower-dimensional representation before constructing the $k$-NN graph. In fact, Fig. 2d shows for four example datasets that the number of dimensions for the PCA was an important determinant of performance. This is because the dimension reduction acts as a smoothener, whose smoothing effect needs to be strong enough to average out uncorrelated noise (small enough target space dimension), but flexible enough to maintain interesting variation (large enough target space dimension).

The latent expression-based transformations (except Normalisr) and the count-based factor analysis models were computationally more

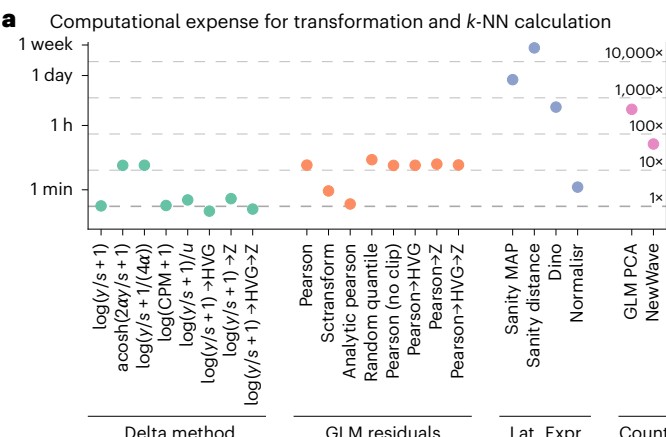

**a** Computational expense for transformation and $k$-NN calculation

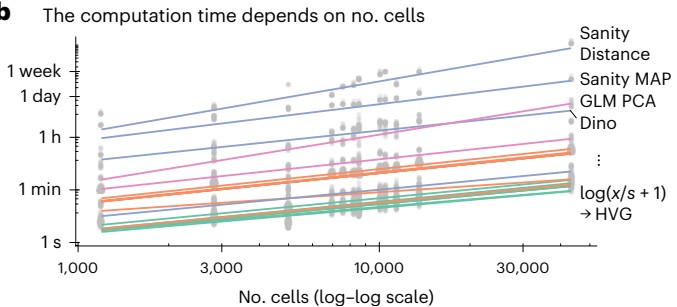

**b** The computation time depends on no. cells

**Fig. 3 | Computational expense. a**, CPU time needed to calculate the transformation and identify the $k$-NNs for the 10x human helper T-cell dataset. The secondary axis shows the duration relative to the shifted logarithm. **b**, Dependence of run time on the number of cells, across datasets, shown on a double-logarithmic scale, with a linear fit. Most transformations have a slope of approximately 1 (scale linearly), whereas Sanity Distance and GLM PCA have a slope > 1.5, which indicates quadratic scaling. Source data are provided.

expensive than the delta method- and residuals-based transformations. Figure 3a shows the CPU times for calculating the transformation and finding the $k$-NN on the 10x human helper T-cell dataset with 10,064 cells. Sanity Distance took particularly long because its distance calculation, which takes into account the uncertainty for the nearest neighbor search, scaled quadratically with the number of cells (Fig. 3b). Across all benchmarks, the computations took 24 years of CPU time, of which the latent expression-based transformations accounted for over 90%. The delta method-based transformations were the fastest, especially if the overdispersion was not estimated from the data. The residuals-based transformations took somewhat more time, except for the analytic approximation of the Pearson residuals, which could be calculated almost as fast as the shifted logarithm. In terms of memory consumption, the delta method-based transformations were most attractive because they retained the sparsity of the data.

In terms of uncovering the latent structure of the datasets, none of the other transformations consistently outperformed the shifted logarithm (Fig. 4a), one of the simplest and oldest approaches. In fact, when followed by PCA dimension reduction to a suitable target dimension, the shifted logarithm performed better than the more complex latent expression-based transformations across all three benchmarks.

We found no evidence that additional post-processing steps (rescaling the output of the shifted logarithm, selecting HVGs or equalizing the variance of all genes using z scoring) improved the results for identifying nearest neighbors (Fig. 4b). Lause et al.[20] and Choudhary and Satija[25] debated on how to best choose the

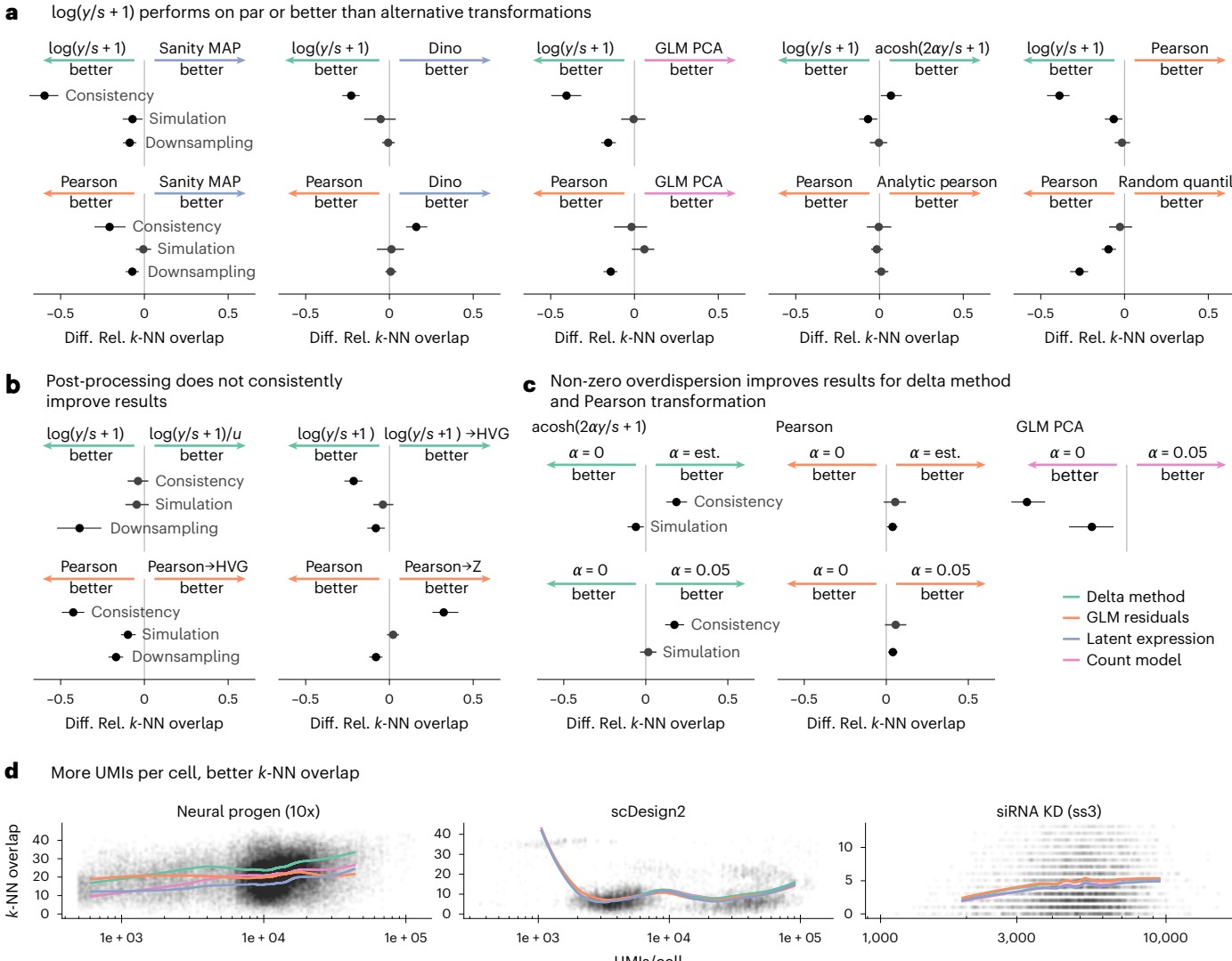

**Fig. 4 | Comparison of selected transformations. a–c**, 95% confidence intervals (based on $n$ = 5 replicates over five (simulation, downsampling) / ten (consistently) independent datasets) of the differences of the relative $k$-NN overlap between selected transformations as shown in Fig. 2. Shifted logarithm and Pearson residuals against a selection of the best performing transformations from the other preprocessing approaches (**a**). Effect of applying various post-processing methods after applying the shifted logarithm or Pearson residuals transformation (**b**). Effect of fixing the overdispersion to 0 or 0.05, or estimating a gene-specific overdispersion from the data (**c**). **d**, Smoothed line plots of the $k$-NN overlap ($y$ axis) as a function of the UMIs per cell ($x$ axis) for the shifted logarithm transformation, the Pearson residuals, Sanity MAP and GLM PCA are colored by the respective transformation approach. Source data are provided.

overdispersion parameter. We found empirically that, for Pearson residuals and the acosh transformation, it is beneficial to have $\alpha > 0$, but saw no clear benefits from estimating this parameter from the input data versus using a generic, fixed value such as 0.05 (Fig. 4c).

Last, we found that with increasing sequencing depth per cell, all methods generally had a better $k$-NN overlap with the ground truth (Fig. 4d). This makes intuitive sense; with higher sequencing depth, the relative size of the sampling noise is reduced. Based on Fig. 1a, we might assume that delta method-based transformations would perform particularly poorly at identifying the neighbors of cells with extreme sequencing depths; yet on three datasets, the shifted logarithm did not perform worse than other transformations for cells with particularly large or small size factors (Fig. 4d). We also considered the performance of the transformations as a function of cluster size (Extended Data Figs. 10); while we saw some interesting variation, we did not find that a single transformation performed consistently better or worse for small clusters.

## Discussion

We compared 22 transformations, conceptually grouped into four basic approaches, for their ability to recover latent structure among the cells. We found that one of the simplest approaches, the shifted logarithm transformation $\log(y/s + y_0)$ with $y_0 = 1$ followed by PCA, performed surprisingly well. We presented theoretical arguments for using the related acosh transformation or an adaptive pseudo-count $y_0 = 1/(4\alpha)$, but our benchmarks showed limited performance benefits for these.

We recommend against using CPM as input for the shifted logarithm. We pointed out that for typical datasets, this amounts to assuming an unrealistically large overdispersion and in our benchmarks this approach performed poorly compared to applying the shifted logarithm to size factor-scaled counts. We also advise against scaling the results of the shifted logarithm by the sum of the transformed values per cell as, for example, suggested by Booeshaghi et al.[19]. In our hands (Extended Data Fig. 1), this additional operation failed to remove

the confounding effect of the sequencing depth (the authors' stated motivation for it) and did not improve the $k$-NN recall performance.

The Pearson residuals-based transformation has attractive theoretical properties and, in our benchmarks, performed similarly well as the shifted logarithm transformation. It stabilizes the variance across all genes and is less sensitive to variations of the size factor (Extended Data Fig. 1b). The analytic approximation suggested by Lause et al.[20] is appealing because it worked as well as the exact Pearson residuals but could be calculated faster. However, as seen in equation (4), the Pearson residuals-based transformation is affine linear when considered as a function per gene and this may be unsatisfactory for genes with a large dynamic range across cells. As an alternative, we considered randomized quantile residuals as a non-linear transformation, but found no performance improvement. This result exemplifies that choosing a transformation for conceptual reasons does not necessarily translate into better downstream analysis results.

The use of the inferred latent expression state as a transformation and count-based latent factor models are appealing because of their biological interpretability and mathematical common sense. In particular, Sanity Distance is appealing because it does not have any tunable parameters; however, all these transformations performed worse than the shifted logarithm with a reasonable range of PCA dimensions in our benchmarks and some of the transformations were exceptionally computationally expensive (for example, the median CPU time of Sanity Distance was 4,500-times longer than for the shifted logarithm).

Our results partially agree and disagree with previous studies. Germain et al.[24] benchmarked many steps of a typical single-cell RNA-seq analysis pipeline, including a comparison of clustering results obtained after different transformations against a priori assigned populations. In line with our findings, they reported that dimension reduction was of great importance. They went on to recommended sctransform (Pearson residuals) based on its good performance on the Zhengmix4eq dataset, which is a mixture of peripheral blood mononuclear cells sorted by surface markers using flow cytometry; however, it is not clear how generalizable this result is and our benchmarks do not support such a singling out of that method. Lause et al.[20] considered the related Zhengmix8eq dataset, into which they implanted a synthetic rare cell type by copying 50 B cells and appending ten genes exclusively expressed in the synthetic cell type. They used $k$-NN classification accuracy of the cell type averaged per cell type (macro F1 score; Fig. 5c of their paper) and averaged over all cells (online version of Fig. 5c). They found a performance benefit for the Pearson residuals over the shifted logarithm with the macro F1 score, but similar performance with regard to overall accuracy. The macro F1 score emphasizes the performance difference for the synthetic cell type, which seems somewhat construed and might not be a good model for most biologically relevant cell type and state differences. Instead of comparing clustering results to discrete cell type assignments, we have focused on the inference of the $k$-NN of each cell, with the expectation that this enables consideration of more subtle latent structures than well-separated, discrete cell types.

Pearson residuals- and delta method-based transformations weight genes differently; for example, Pearson residuals put more weight on lowly expressed genes than the delta method (Fig. 1b). This can lead to different downstream results, but our benchmarks did not indicate that any particular weighting is generally better; only that the delta method-based transformation produced more consistent results on the 10x datasets.

We did not evaluate the impact of alternative size factor estimators. We also did not consider how suitable a transformation is for marker gene selection, because we are not aware of a suitable metric to determine success, as the utility of a marker gene hinges on its biological interpretability. For a recent effort to compare different marker gene selection methods, see Pullin and McCarthy[26].

Considerable research effort has been invested in the area of preprocessing methods for single-cell RNA-seq data. To our surprise,

the shifted logarithm still performs among the best. Our bottom-line performance benchmark highlights current limitations of theoretical analysis of preprocessing methods, but also the utility of lower-dimensional embeddings of the transformed count matrix to reduce noise and increase fidelity. Interesting open questions include choosing among the many possible embedding methods and number of latent dimensions.

## Online content

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

## Methods

### The delta method

The delta method is a way to find the s.d. of a transformed random variable.

If we apply a differentiable function $g$ to a random variable $X$ with mean $\mu$, the s.d. of the transformed random variable $g(X)$ can be approximated by

$$\mathbb{S}\mathrm{d}[g(X)] \approx a\,\mathbb{S}\mathrm{d}[X],$$

where $a = |g'(\mu)|$ is the slope of $g$ at $\mu$.

Now consider a set of random variables $X_1, X_2, \ldots$ whose variances and means are related through some function $v$, that is, $\mathbb{V}\mathrm{ar}[X_i] = v(\mu_i)$, or equivalently $\mathbb{S}\mathrm{d}[X_i] = \sqrt{v(\mu_i)}$. Then we can find a variance-stabilizing transformation $g$ by requiring constant s.d., $\mathbb{S}\mathrm{d}[g(X_i)] = \text{const.}$, which using the above approximation becomes

$$g'(\mu) = \frac{\text{const.}}{\sqrt{v(\mu)}},$$

and can be solved by integration.

### Transformations

We compared 22 transformations that can be grouped into four approaches.

The delta method-based transformations were: the shifted logarithm ($\log(y/s + 1)$); the acosh transformation ($\mathrm{acosh}(2\alpha y/s + 1)$); the shifted logarithm with pseudo-count dependent on the overdispersion ($\log(y/s + 1/(4\alpha))$); the shifted logarithm with CPM ($\log(\mathrm{CPM} + 1)$); the shifted logarithm with subsequent size normalization as suggested by Booeshaghi et al.[19] ($x_{gc}/u_c$, where $x_{gc} = \log(y_{gc}/s_c + 1)$ and $u_c = \sum_g x_{gc}$); the shifted logarithm with subsequent HVG selection ($\log(y/s + 1) \rightarrow$ HVG); the shifted logarithm with subsequent $z$ scoring per gene ($\log(y/s + 1) \rightarrow$ Z); and the shifted logarithm with subsequent highly variable gene selection and $z$ scoring per gene ($\log(y/s + 1) \rightarrow$ HVG $\rightarrow$ Z). For all composite transformations, we first calculated the variance-stabilizing transformation, then chose the HVGs and used the results without recalculating the variance-stabilizing transformation.

To retain the sparsity of the output also if the pseudo-count $y_0 \neq 1$, transformGamPoi uses the relation

$$\log\left(\frac{y}{s} + y_0\right) = \log\left(\frac{y}{y_0\,s} + 1\right) + \log y_0. \tag{6}$$

Subtracting the constant $\log y_0$ from this expression does not affect its variance-stabilizing properties, but has the desirable effect that data points with $y = 0$ are mapped to 0.

The residuals-based transformations were: Pearson residuals implemented with the transformGamPoi package where each residual is clipped to be within $\pm\sqrt{\text{no. cells}}$, as suggested by Hafemeister and Satija[13] (Pearson); Pearson residuals with clipping and additional heuristics implemented by sctransform v.2, an analytic approximation to the Pearson residuals with clipping suggested by Lause et al.[20] (Analytic Pearson); randomized quantile residuals implemented by transformGamPoi (Random Quantile); Pearson residuals without clipping implemented by transformGamPoi (Pearson (no clip)); Pearson residuals with clipping and subsequent HVG selection (Pearson $\rightarrow$ HVG); Pearson residuals with clipping and subsequent $z$ scoring per gene (Pearson $\rightarrow$ Z); and Pearson residuals with clipping and subsequent HVG selection and $z$ scoring per gene (Pearson $\rightarrow$ HVG $\rightarrow$ Z). For each composite Pearson residual transformation (with HVG and/or $z$ scoring), we used the transformGamPoi implementation.

The latent expression-based transformations were: Sanity with point estimates for the latent expression (Sanity MAP) and with

calculation of all cell-by-cell distances taking into account uncertainty provided by the posteriors (Sanity Distance); Dino as provided in the corresponding R package; and Normalisr with variance normalization, implemented in Python, which we called from R using the reticulate package.

The count-based factor analysis models were: GLM PCA using the Poisson model and the gamma-Poisson model with $\alpha = 0.05$. In the figures, we show the results for the Poisson model unless otherwise indicated. We used the avagrad optimizer. We ran NewWave with 100 genes for the mini-batch overdispersion estimation.

For the delta method-based transformations and the residuals-based transformations calculated with the transformGamPoi package, we calculated the size factor $s$ using equation (3).

We defined HVGs as the 1,000 most variable genes based on the variance of the transformed data.

For $z$ scoring, we took the transformed values $x_{gc} = g(y_{gc})$ and computed $z_{gc} = \frac{x_{gc} - \mathrm{mean}(\boldsymbol{x}_g)}{\sqrt{\mathrm{var}(\boldsymbol{x}_g)}}$, where mean and variance are the empirical mean and variance taken across cells.

In the overview figures (Figs. 2–4), we use a gene-specific overdispersion estimate for all residuals-based transformations and for the delta method-based transformations, which can handle a custom overdispersion; for GLM PCA, we use $\alpha = 0$, because these settings worked best for the respective transformations. The latent expression-based transformations and NewWave do not support custom overdispersion settings.

### Conceptual differences

For the visualization of the residual structure after adjusting for the varying size factors, we chose a control dataset of a homogeneous RNA solution encapsulated in droplets[21]. We filtered out RNAs that were all zero and plotted the first two principal components. Where applicable, we used gene-specific overdispersion estimates. For visualizing the results of Sanity Distance, instead of the PCA, we used multidimensional scaling of the cell-by-cell distance matrix using R's cmdscale function. We calculated the canonical correlation using R's cancor function on the size factors and the first ten dimensions from PCA and multidimensional scaling.

The plots of the mean–variance relationship are based on the 10x human hematopoietic cell dataset[27]. Where applicable, we used gene-specific overdispersion estimates. The panel of Sanity Distance shows the variance of samples drawn from a normal distribution using the inferred mean and s.d.

For the mouse lung dataset[28], we filtered out cells with extreme size factors ($0.1s_{\mathrm{median}} < s_c < 10s_{\mathrm{median}}$, where $s_{\mathrm{median}}$ is the median size factor). We also removed cells that did not pass the scran quality control criterion regarding the fraction of reads assigned to mitochondrial genes. To account for the fact that some transformations share information across genes, we applied all transformations to the 100 most highly expressed genes and three genes (*Sftpc*, *Scgb1a1* and *Ear2*) known to be differentially expressed in some cell types according to the assignment from the original publication.

### Benchmarks

The benchmarks were executed using a custom work scheduler for slurm written in R on CentOS7 and R 4.1.2 with Bioconductor v.3.14. The set of R packages used in the benchmark with exact version information was stored using the renv package and is available from the GitHub repository.

***k*-NN identification and dimensionality reduction.** To calculate the PCA, we used the irlba package. To infer the $k$-NN, we used annoy, which implements an approximate nearest neighbor search algorithm. To calculate $t$-distributed stochastic neighbor embeddings (tSNEs), which we only used for visualization, we used the Rtsne package on

data normalized with the shifted logarithm with a pseudo-count of 1.

**Consistency benchmark.** We downloaded ten single-cell datasets listed in GEO browser after searching for the term mtx on 14 October 2021. All datasets are listed in the Data Availability section. To measure the consistency of the transformations, we randomly assigned each gene to one of two groups and processed the two resulting data subsets separately. We calculated the consistency as the mean overlap of the $k$-NN for all cells.

**Simulation benchmark.** We used five frameworks to simulate single-cell counts in R: we ran dyngen[29] using a consecutive bifurcating mode and the default parameters otherwise. We ran muscat[30] with four clusters, a default of 30% differentially expressed genes with an average log fold change of 2 and a decreasing relative fraction of log fold changes per cluster. We ran scDesign2 (ref. [31]) with the 10x human hematopoietic cell dataset as the reference input with a copula model and a gamma-Poisson marginal distribution. We simulated the Random Walk by translating the MATLAB code of Breda et al.[14] to R and using the data by Baron et al.[32] as a reference. For the Linear Walk, we adapted the Random Walk simulation and, instead of following a Random Walk for each branch, we interpolated the cells linearly between a random start and end point. For both benchmarks, we used a small non-zero overdispersion of $\alpha = 0.01$ to mimic real data.

With each simulation framework, we knew which cells were the $k$-NNs to each other. We calculated the overlap as the mean overlap of this ground truth with the inferred nearest neighbors on the simulated counts for all cells. Furthermore, we calculated the ARI and AMI by clustering the ground truth and the transformed values with the graph-based walktrap clustering algorithm from the igraph package.

**Downsampling benchmark.** We searched the literature for single-cell datasets with high sequencing depth and found five (one from mcSCRB, four from Smart-seq3) that had a sequencing depth of more than 50,000 UMIs per cell on average. We defined reliable nearest neighbors as the set of $k$-NNs of a cell that were identified with all 22 transformations on the deeply sequenced data (excluding the two negative controls). We used the downsampleMatrix function from the scuttle package to reduce the number of counts per cell to approximately 5,000, a typical value for 10x data. We considered only one setting for the overdispersion per transformation (instead of allowing multiple overdispersion settings for some transformations as in the other benchmarks). We ran all transformations that supported the setting, with a gene-specific overdispersion estimate (except GLM PCA, which performed better with an overdispersion fixed to 0). Finally, we computed the mean overlap between the $k$-NNs identified on the downsampled data with the set of reliable nearest neighbors for all cells with more than one reliable nearest neighbor.

**$k$-NN overlap.** For all three benchmarks, we calculated overlaps between pairs of $k$-NN graphs. Denoting their no. cell × no. cell adjacency matrices (a matrix of zeros and ones, where an entry is is one if a cell $d$ is among the $k$-NNs of cell $c$) by $N^1$ and $N^2$, we defined their overlap as

$$\frac{1}{\text{No. cells}} \sum_{c,d=1}^{\text{No. cells}} N^1_{cd} N^2_{cd}. \tag{7}$$

**Reporting summary**

Further information on research design is available in the Nature Portfolio Reporting Summary linked to this article.

## Data availability
All datasets are freely available. Source data have been provided for Figs. 1–4 and Extended Data Figs. 1–3 and 5–10 (refs. 21,27,28,32–44). Source data are provided with this paper.

## Code availability
An R package that implements the delta method- and residuals-based variance-stabilizing transformations is available at bioconductor.org/packages/transformGamPoi/. The code to reproduce the analysis and generate the figures is available at github.com/const-ae/transformGamPoi-Paper and stored permanently with Zenodo[45]. We provide an interactive website to explore the benchmark results at shiny-portal.embl.de/shinyapps/app/08_single-cell_transformation_benchmark.

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

## Acknowledgements

We thank S. Anders for extensive discussions about variance-stabilizing transformations and how to benchmark preprocessing methods. We thank the three anonymous reviewers, E. van Nimwegen and D. Kobak, whose feedback on an earlier version helped to improve the paper. This work has been supported by the EMBL International PhD Program (C.A.E.), by the German Federal Ministry of Education and Research (CompLS project SIMONA under grant agreement no. 031L0263A) (C.A.E. and W.H.) and the European Research Council (Synergy Grant DECODE under grant agreement no. 810296) (C.A.E. and W.H.).

## Author contributions

C.A.E. and W.H. conceived the idea for the study and wrote the final paper. C.A.E. performed all computations with feedback from W.H.

## Funding

## Competing interests

The authors declare no competing interests.

## Additional information

**Extended data** is available for this paper at https://doi.org/10.1038/s41592-023-01814-1.

**Correspondence and requests for materials** should be addressed to Constantin Ahlmann-Eltze.

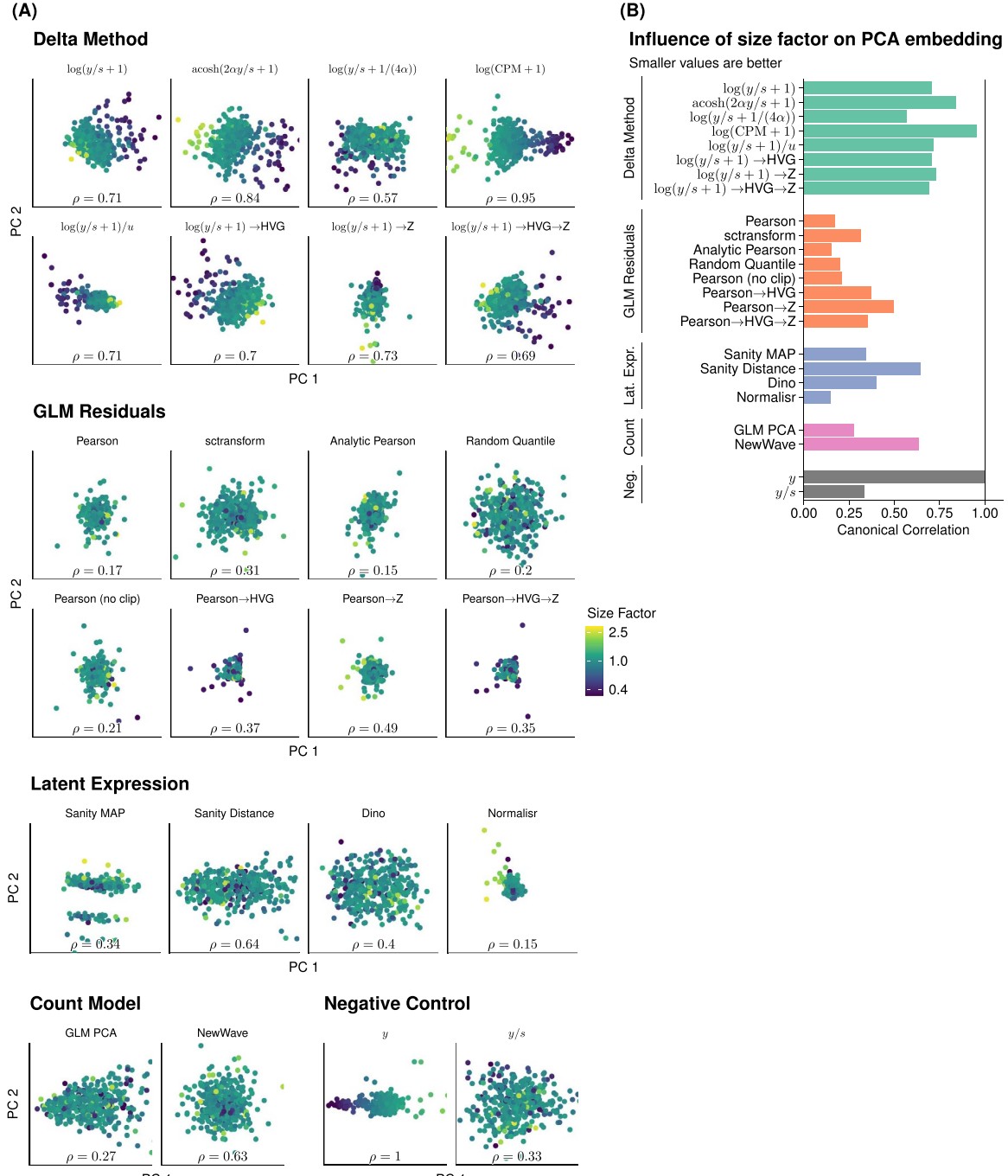

**Extended Data Fig. 1 | Confounding effect of size factors on PCA embedding of a homogeneous dataset.** (A) Scatter-plots of the first two principal components of the transformed data colored by the sequencing depth (expressed as a normalized size factor on a logarithmic scale) per cell. The data are from droplets that encapsulate a homogeneous RNA solution and thus the only variation is due to technical factors like sequencing depth[21]. The annotation at the bottom of the plot shows the canonical correlation coefficient $\rho$[46] between the size factor and the first ten principal components. A lower canonical correlation that the variance-stabilizing transformation more successfully adjusts for the varying size factors; a canonical correlation of $\rho = 1$ means that the ordering of the cells along some direction in the first 10 PCs is entirely determined by the size factor. (B) Collection of the canonical correlations from the annotations of each panel in A displayed as a bar chart for easy visual comparison.

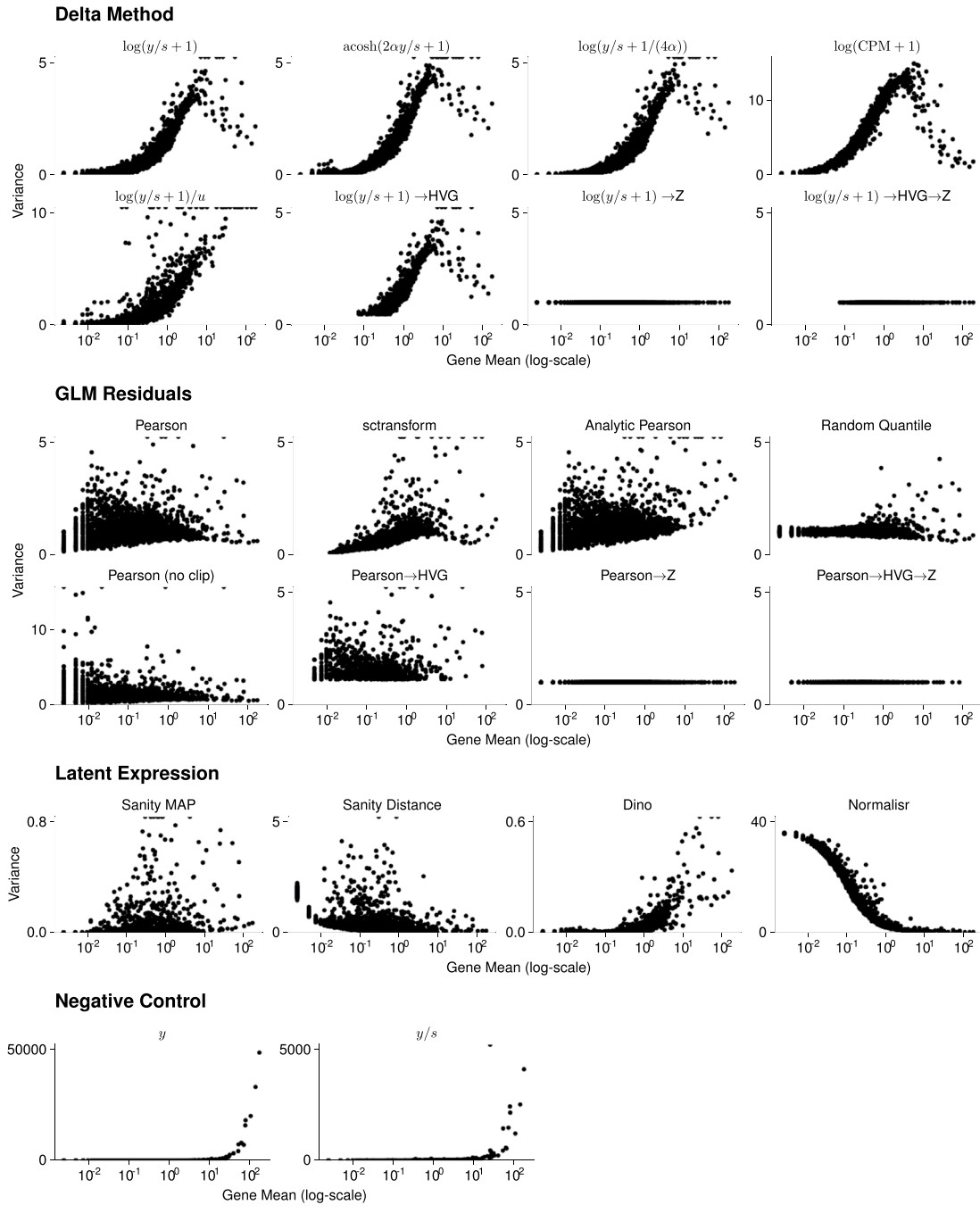

**Extended Data Fig. 2 | Mean-variance relations per gene for a 10x hematopoietic cell dataset.** Scatter-plots of the variance per gene after applying the variance-stabilizing transformation against the means of the 10x human hematopoietic cell dataset subset to 400 cells and 5000 genes.

Note that the scale of the y axis differs for the raw counts, log(CPM + 1), log($y/s$ + 1)/$u$, Pearson (no clip), Sanity MAP, Dino and Normalisr for esthetic purposes. Points that exceed the y axis scale are drawn on the top of each facet.

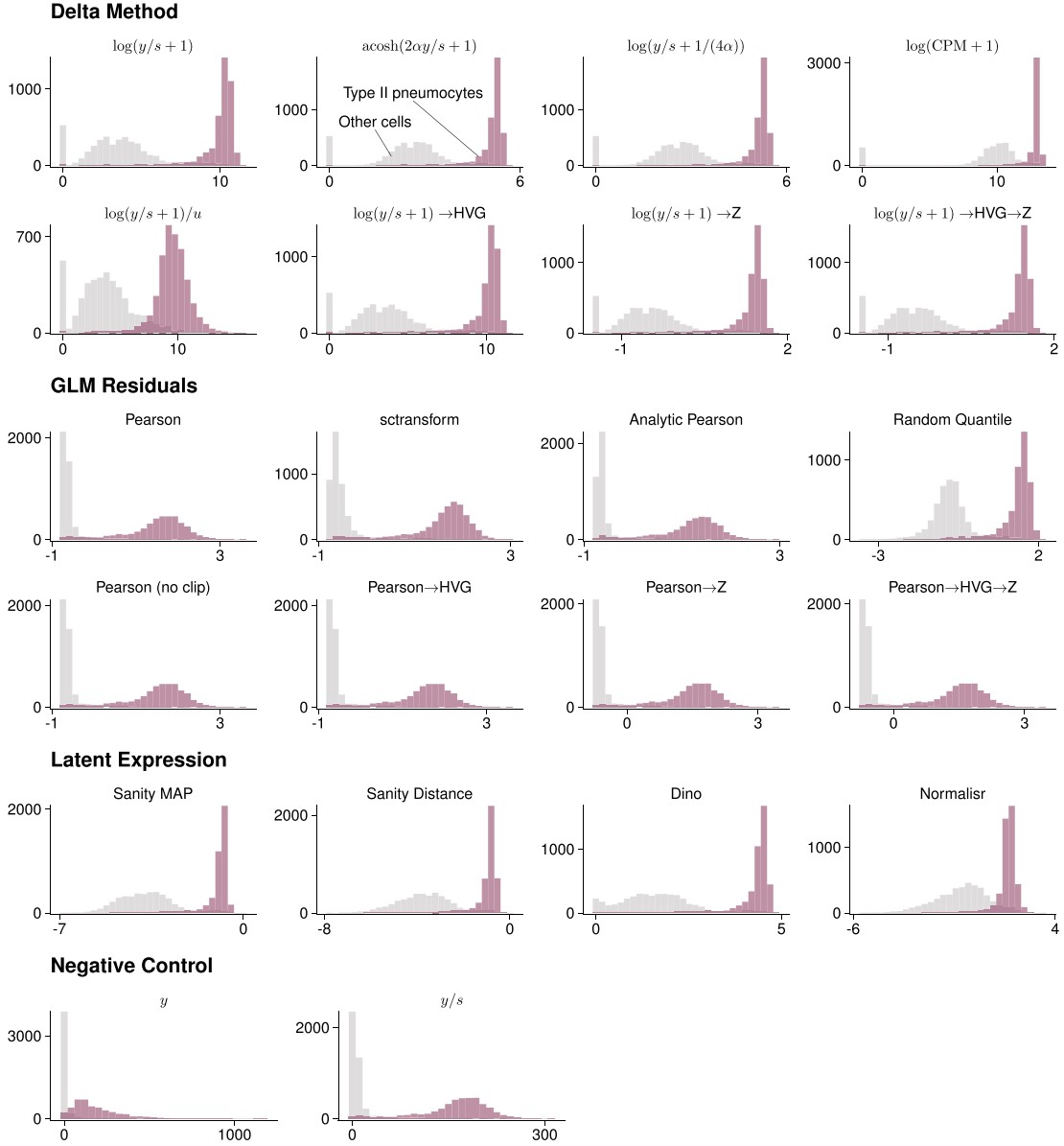

**Extended Data Fig. 3 | Histograms of the transformed values for a gene with a bimodal expression pattern.** Counts from cells identified as type II pneumocytes are shown in purple and a matching number of counts from all other cell types are shown in gray.

# Construction of Randomized Quantile Residuals

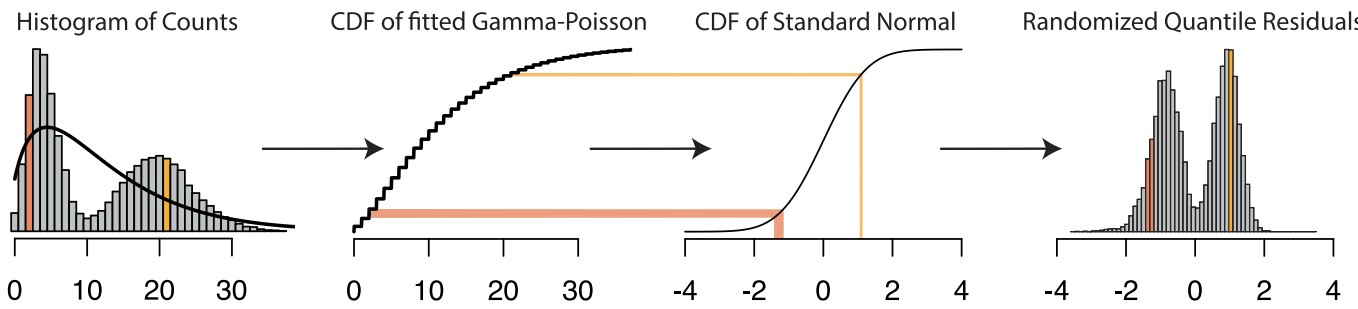

**Extended Data Fig. 4 | Schematic representation of how randomized quantile residuals are constructed.** In the first step, a Gamma-Poisson distribution (black line) is fitted to the observed counts. Then, the quantiles of the Gamma-Poisson distribution are matched with the quantiles of a standard normal distribution by comparing their respective cumulative density functions (CDFs). This obtains a mapping from the raw count scale to a new, continuous scale. The two colored bars (orange for $y = 2$, yellow for $y = 21$) exemplify this mapping. The non-linear nature of the CDFs ensures that small counts are mapped to a broader range than large counts. This helps to stabilize the variance on the residual scale. Furthermore, the randomization within the mapping sidesteps the discrete nature of the counts.

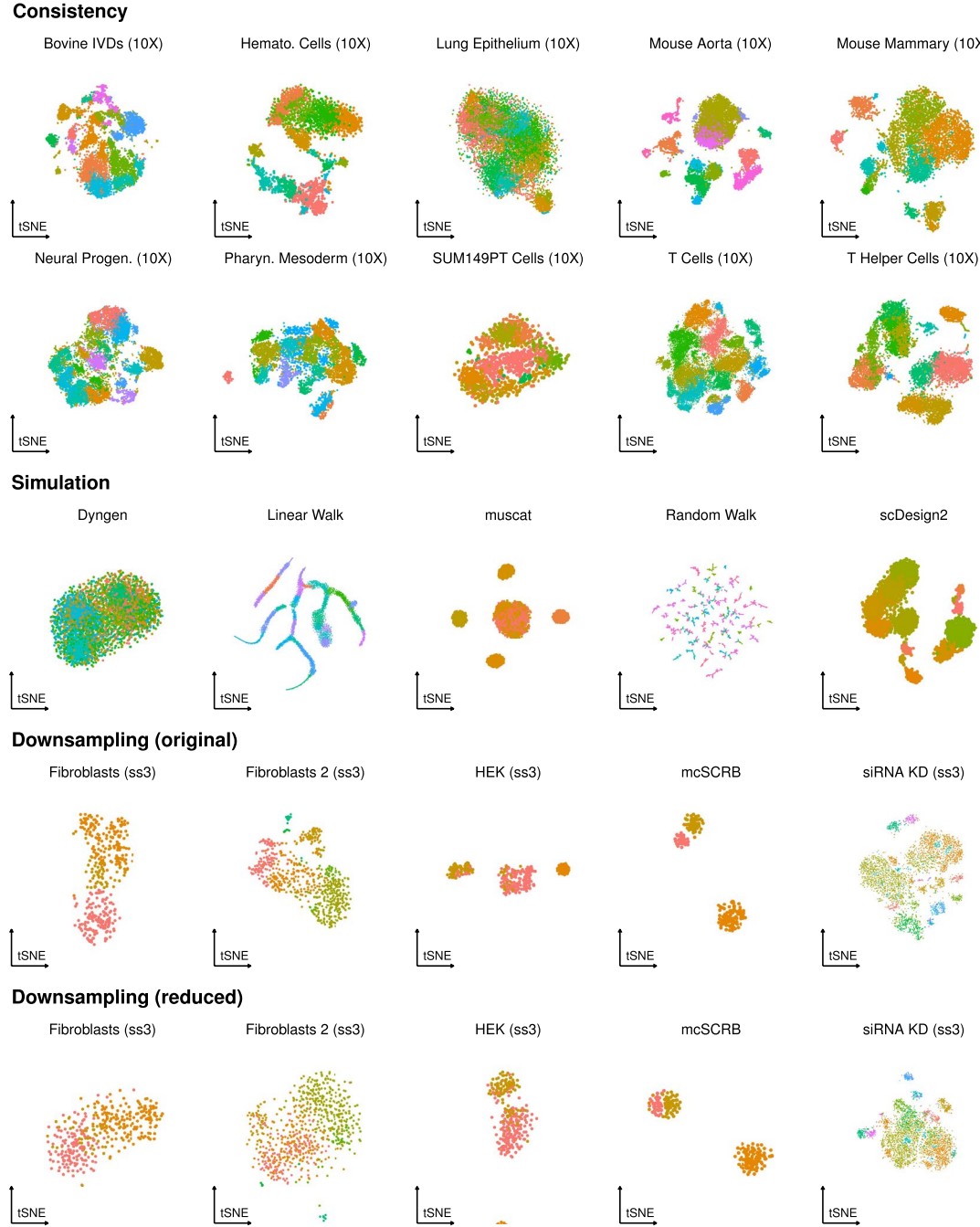

**Extended Data Fig. 5 | tSNE plots of each dataset used for the benchmarks.** The cells are colored by clustering using the walktrap clustering algorithm. For the consistency data we clustered the counts after transformation with the shifted logarithm. For the simulation data, we clustered the ground truth. For the downsampling data, we clustered the deeply sequenced data after transformation with the shifted logarithm.

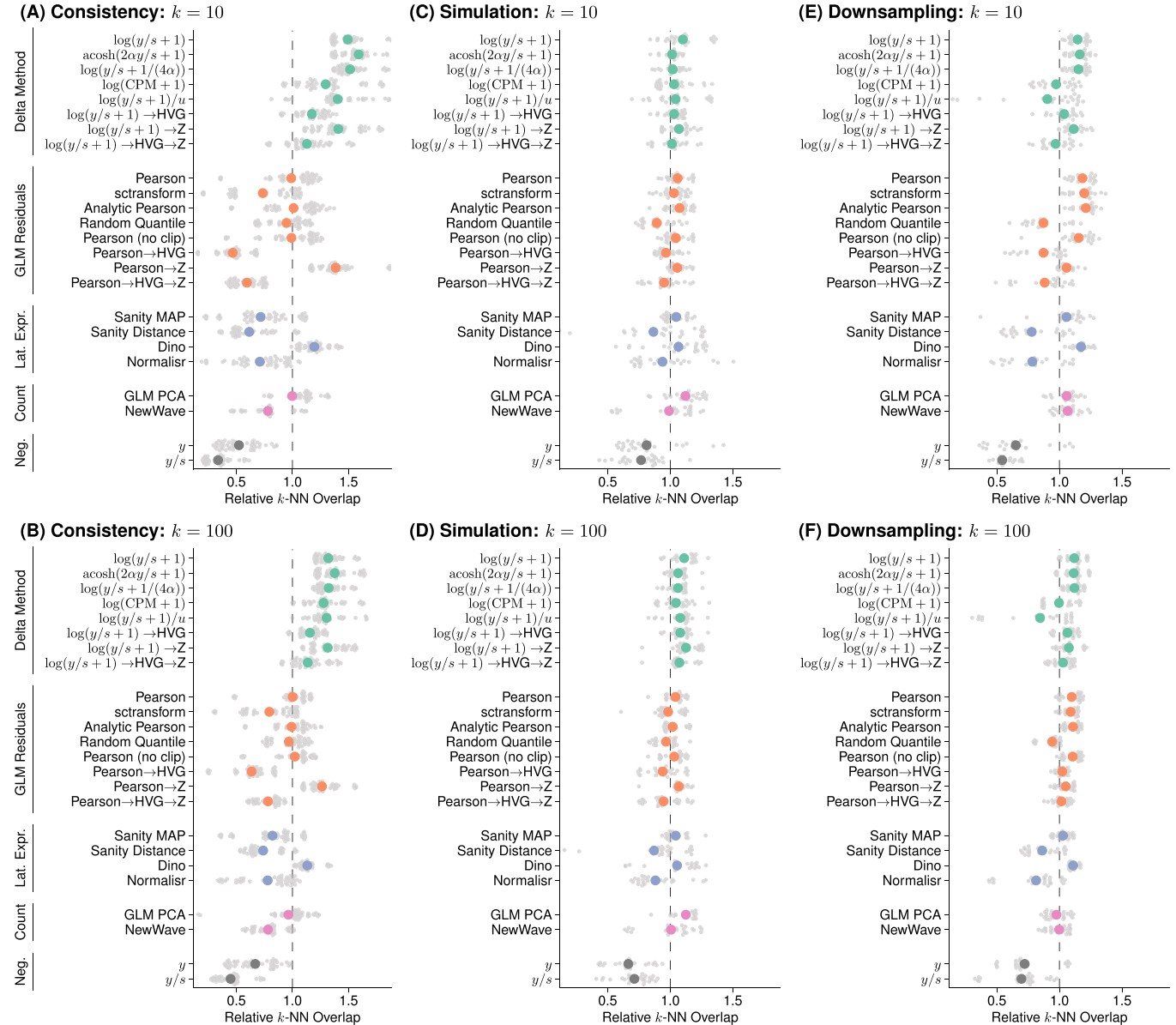

**Extended Data Fig. 6 | Benchmark results for 10 and 100 nearest neighbors.**
Plot of the aggregate results of the consistency (A, B), simulation (C, D) and downsampling benchmarks (E, F) for k = 10 and k = 100, respectively. The results for each dataset are broad to a common scale by normalizing to the mean *k* nearest neighbor overlap per dataset. The colored points show the averages across the datasets, each with 5 replicate random data splits (small, gray points).

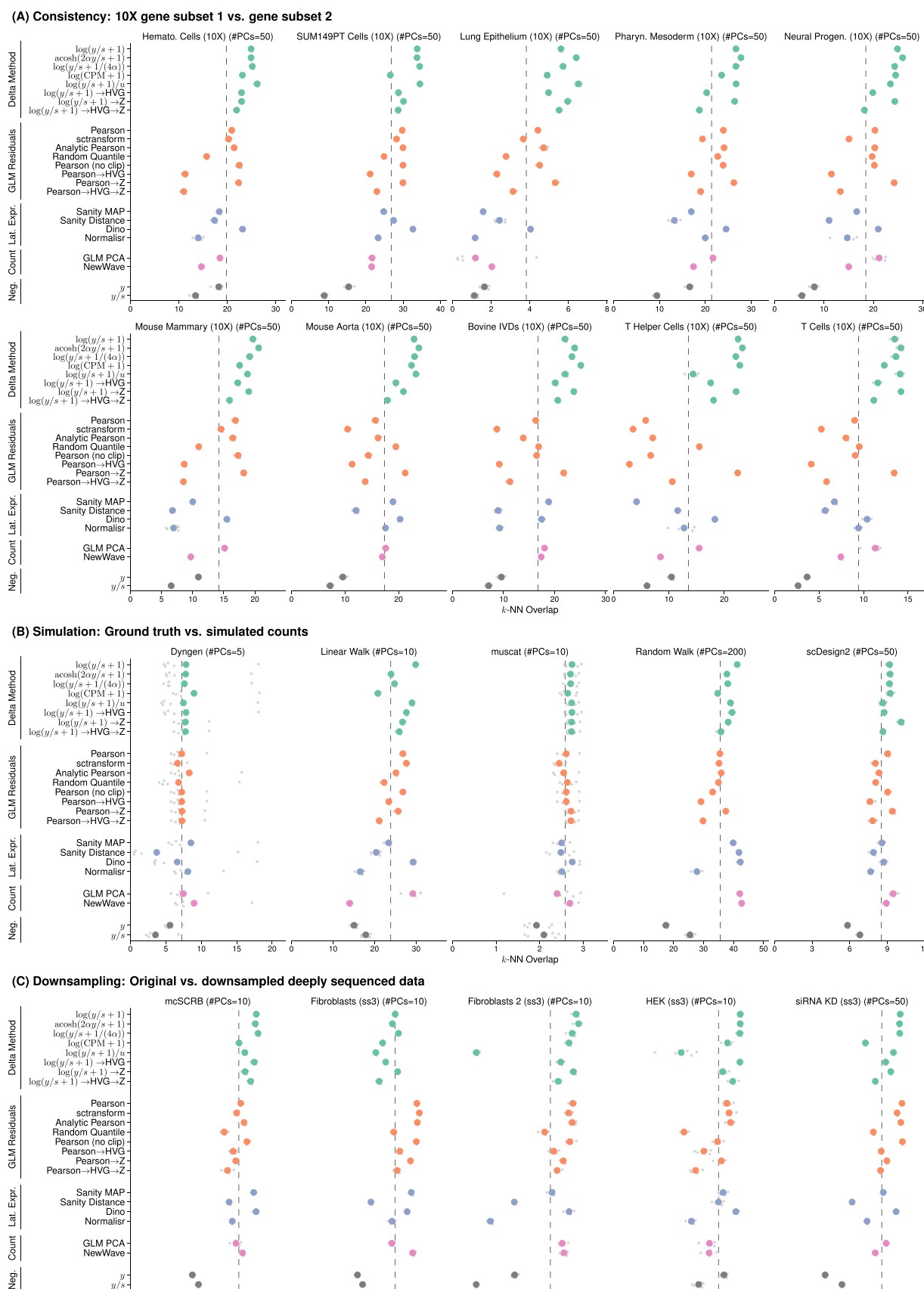

**Extended Data Fig. 7 | The unaggregated benchmark results.** The unaggregated results from the consistency (A), simulation (B) and downsampling benchmarks (C) for $k = 50$. The gray points show the raw results from the five replicates per dataset; the colored points show their mean. The dashed vertical line indicates the mean $k$-NN overlap per dataset and is the reference used to aggregated the results as shown in Fig. 2A-C. The title of each facet indicates the number of dimensions used for the PCA per dataset, which we chose based on the complexity of the dataset.

**(A) Pairwise $k$-NN overlap**

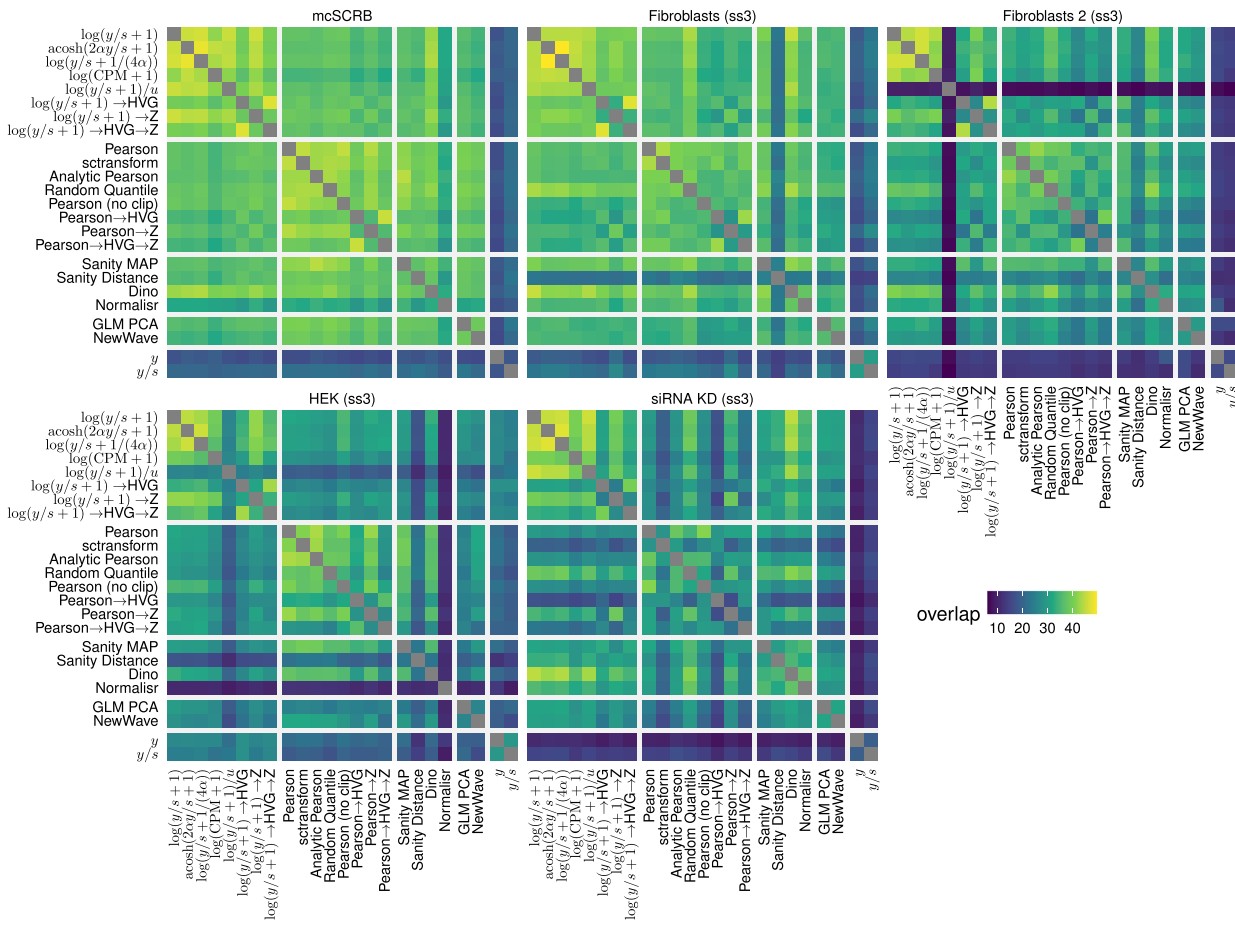

**(B) Histogram of reliable nearest neighbors per cell considering all transformations**

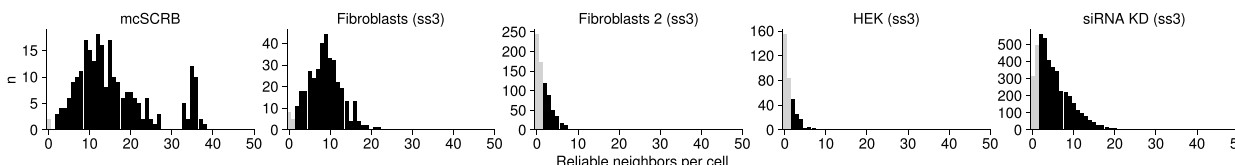

**(C) Histogram of reliable nearest neighbors per cell considering only the top two transformations per approach**

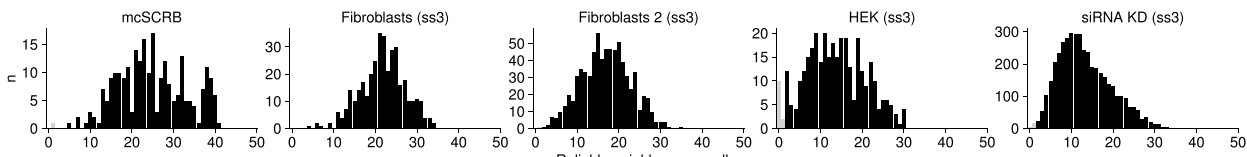

**(D) Downsamping results considering only the top two transformations per approach**

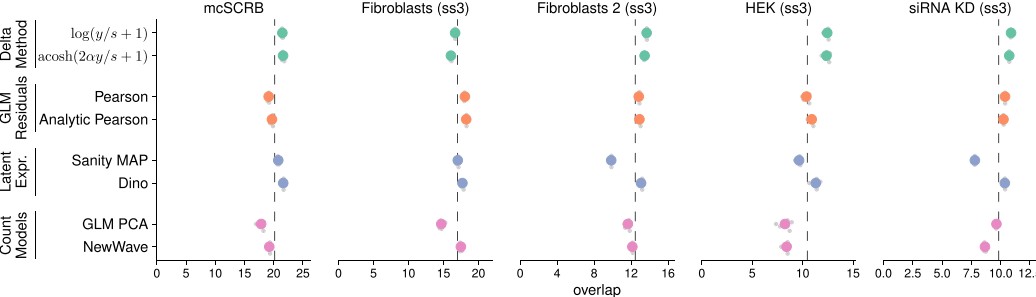

**Extended Data Fig. 8 | See next page for caption.**

**Extended Data Fig. 8 | Inference of the reliable nearest neighbors for the deeply sequenced datasets.** (A) Heatmaps of the average *k*-NN overlap for all transformation pairs. (B) Histograms of the number of reliable neighbors per cell (that is, the neighbors among the 50 *k*-NN that were identified by all 22 transformations). The dark shaded bars show the cells that were used to calculate the overlap with the downsampled version of the data in Extended Data Fig. 7C.

(C) Histograms of the number of reliable neighbors per cell only considering the two two transformations per approach (that is, the neighbors among the 50 *k*-NN that were identified by 8 transformations listed in (D)). (D) The unaggregated results for the downsampling benchmarks using the same settings as in Extended Data Fig. 7C.

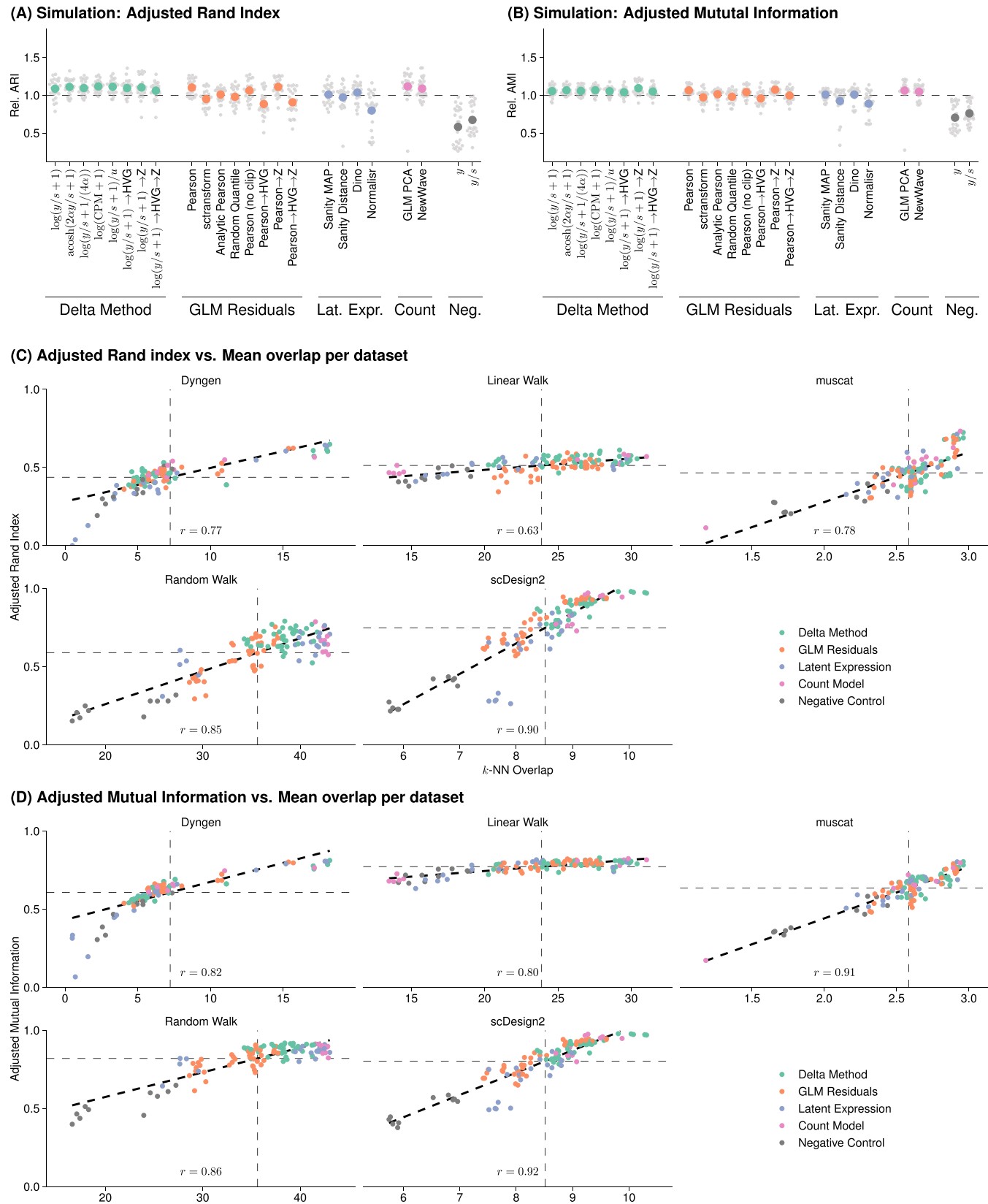

**Extended Data Fig. 9 | Results of the simulation benchmark in terms of cluster similarity.** Plots of the results using the adjusted Rand index (A) and the adjusted mutual information (B) instead of the *k*-NN overlap. (C-D) Scatter-plots facetted by simulation framework that compares the results for the *k*-NN overlap with the adjusted Rand index and adjusted mutual information, respectively. Each point is one replicate for the transformation results of that dataset colored by the transformation approach. The black dashed line shows the linear fit and the number at the bottom of each plot is the correlation coefficient. The horizontal dashed line is the mean ARI / AMI that is used for forming the relative performance in (A) and (B). The vertical dashed line is the mean *k*-NN overlap and corresponds to the vertical dashed line in Extended Data Fig. 7B.

**(A)** $k$-**NN overlap stratified by cluster size for Neural Progen. (10X)**

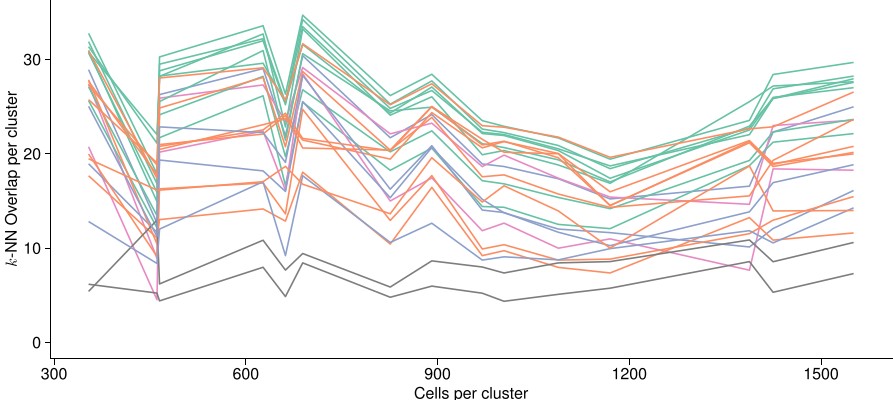

**(B)** $k$-**NN overlap stratified by cluster size for scDesign2**

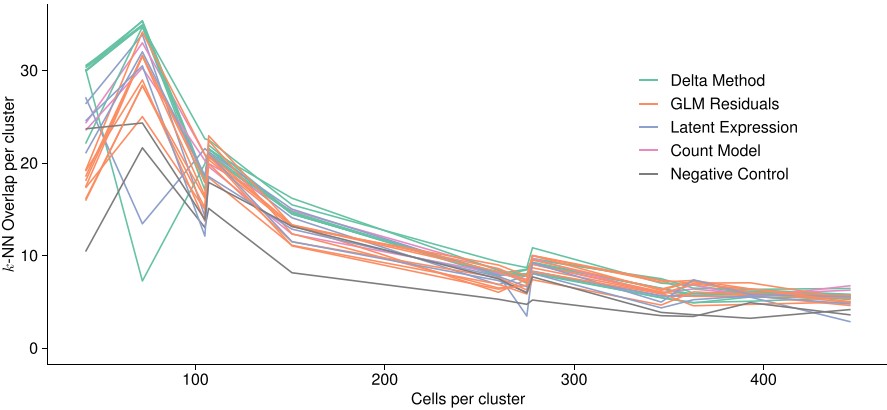

**(C)** $k$-**NN overlap stratified by cluster size for siRNA KD (ss3)**

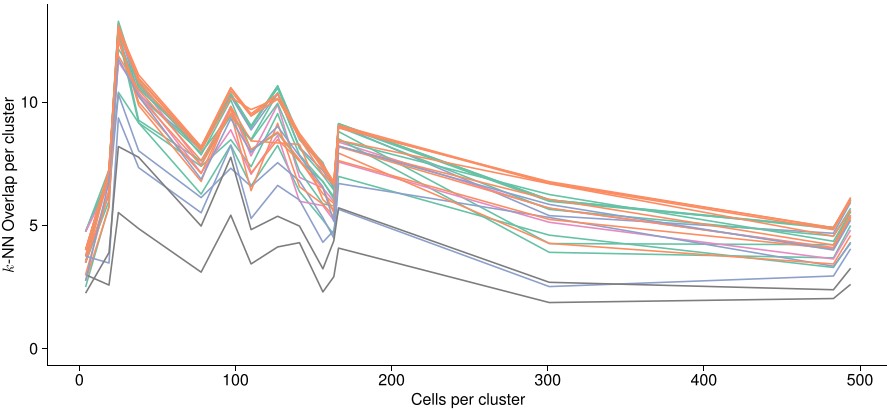

**Extended Data Fig. 10 | $k$-NN overlap as a function of cluster size.** The datasets were clustered using walktrap clustering. Extended Data Fig. 5 shows the cells colored by cluster assignment for all datasets. (A) $k$-NN overlap of the two halves of the *human neural progenitor* dataset stratified by cluster. (B) $k$-NN overlap with the ground truth for the *scDesign*2 simulation stratified by cluster. (C) $k$-NN overlap with the deeply sequenced data for the *siRNA knockdown* dataset stratified by cluster.

# Reporting Summary

## Statistics

For all statistical analyses, confirm that the following items are present in the figure legend, table legend, main text, or Methods section.

| n/a | Confirmed | |
|---|---|---|
| ☒ | ☐ | The exact sample size (*n*) for each experimental group/condition, given as a discrete number and unit of measurement |
| ☒ | ☐ | A statement on whether measurements were taken from distinct samples or whether the same sample was measured repeatedly |
| ☒ | ☐ | The statistical test(s) used AND whether they are one- or two-sided<br>*Only common tests should be described solely by name; describe more complex techniques in the Methods section.* |
| ☒ | ☐ | A description of all covariates tested |
| ☒ | ☐ | A description of any assumptions or corrections, such as tests of normality and adjustment for multiple comparisons |
| ☐ | ☒ | A full description of the statistical parameters including central tendency (e.g. means) or other basic estimates (e.g. regression coefficient) AND variation (e.g. standard deviation) or associated estimates of uncertainty (e.g. confidence intervals) |
| ☒ | ☐ | For null hypothesis testing, the test statistic (e.g. *F*, *t*, *r*) with confidence intervals, effect sizes, degrees of freedom and *P* value noted<br>*Give P values as exact values whenever suitable.* |
| ☒ | ☐ | For Bayesian analysis, information on the choice of priors and Markov chain Monte Carlo settings |
| ☒ | ☐ | For hierarchical and complex designs, identification of the appropriate level for tests and full reporting of outcomes |
| ☒ | ☐ | Estimates of effect sizes (e.g. Cohen's *d*, Pearson's *r*), indicating how they were calculated |

*Our web collection on statistics for biologists contains articles on many of the points above.*

## Software and code

Policy information about availability of computer code

| Data collection | No additional software tools were used for data collection. |
|---|---|
| Data analysis | All code (incl. simulations and data analysis) to reproduce the data analysis is available in the GitHub repository https://github.com/const-ae/transformGamPoi-Paper and stored permanently with Zenodo (https://doi.org/10.5281/zenodo.7504146). The analysis was run using R version 4.1.1. We implemented a package to conduct many of the discussed transformation and made it available on Bioconductor (https://bioconductor.org/packages/transformGamPoi/) in version 3.13.<br><br>We provide an interactive website to explore the benchmark results at https://shiny-portal.embl.de/shinyapps/app/08_single-cell_transformation_benchmark. |

For manuscripts utilizing custom algorithms or software that are central to the research but not yet described in published literature, software must be made available to editors and reviewers. We strongly encourage code deposition in a community repository (e.g. GitHub). See the Nature Portfolio guidelines for submitting code & software for further information.

## Data

Policy information about availability of data

 All manuscripts must include a data availability statement. This statement should provide the following information, where applicable:

- Accession codes, unique identifiers, or web links for publicly available datasets
- A description of any restrictions on data availability
- For clinical datasets or third party data, please ensure that the statement adheres to our policy

| | |
|---|---|
| Droplet encapsulated RNA | Svensson2017 (CalTech Data Repo entry https://data.caltech.edu/records/1264) |
| Human hematopoietic cells | Bulaeva 2020 (GEO https://www.ncbi.nlm.nih.gov/geo/query/acc.cgi?acc=GSE130931) |
| Mouse lung | Angelidis 2019 (GEO https://www.ncbi.nlm.nih.gov/geo/query/acc.cgi?acc=GSE124872) |
| Human hematopoietic cells | Bulaeva 2020 (GEO https://www.ncbi.nlm.nih.gov/geo/query/acc.cgi?acc=GSE130931) |
| SUM149PT cell line | No corresponding publication (GEO https://www.ncbi.nlm.nih.gov/geo/query/acc.cgi?acc=GSE142647) |
| Human lung epithelium | Kathiriya 2022 (GEO https://www.ncbi.nlm.nih.gov/geo/query/acc.cgi?acc=GSE150068) |
| Mouse pharyngeal mesoderm | Nomaru 2021 (GEO https://www.ncbi.nlm.nih.gov/geo/query/acc.cgi?acc=GSE158941) |
| Human neural progenitor cells | DeSantis 2021 (GEO https://www.ncbi.nlm.nih.gov/geo/query/acc.cgi?acc=GSE163505) |
| Mouse mammary | Pal 2021 (GEO https://www.ncbi.nlm.nih.gov/geo/query/acc.cgi?acc=GSE164017) |
| Mouse aorta | Porritt 2021 (GEO https://www.ncbi.nlm.nih.gov/geo/query/acc.cgi?acc=GSE178765) |
| Bovine intervertebral discs (IVDs) | Panebianco 2021 (GEO https://www.ncbi.nlm.nih.gov/geo/query/acc.cgi?acc=GSE179714) |
| Human T helper cells | Qian 2021 (GEO https://www.ncbi.nlm.nih.gov/geo/query/acc.cgi?acc=GSE179831) |
| Human T cells | Lu 2021 (GEO https://www.ncbi.nlm.nih.gov/geo/query/acc.cgi?acc=GSE184806) |
| Human pancreas | Baron 2016 (BioC package https://doi.org/doi:10.18129/B9.bioc.scRNAseq) |
| JM8 cells | Bagnoli 2018 (GEO https://www.ncbi.nlm.nih.gov/geo/query/acc.cgi?acc=GSE103568) |
| HEK cells | Hagemann 2020 (ArrayExpress https://www.ebi.ac.uk/arrayexpress/experiments/E-MTAB-8735/) |
| Fibroblasts (1) | Hagemann 2020 (ArrayExpress https://www.ebi.ac.uk/arrayexpress/experiments/E-MTAB-8735/) |
| Fibroblasts (2) | Larsson 2021 (ArrayExpress https://www.ebi.ac.uk/arrayexpress/experiments/E-MTAB-10148/) |
| siRNA Knockdown (KD) | Johnsson 2022 (Github https://github.com/sandberg-lab/lncRNAs_bursting/tree/main/data) |

## Human research participants

Policy information about studies involving human research participants and Sex and Gender in Research.

| | |
|---|---|
| Reporting on sex and gender | Not applicable |
| Population characteristics | Not applicable |
| Recruitment | Not applicable |
| Ethics oversight | Not applicable |

Note that full information on the approval of the study protocol must also be provided in the manuscript.

# Field-specific reporting

Please select the one below that is the best fit for your research. If you are not sure, read the appropriate sections before making your selection.

☒ Life sciences      ☐ Behavioural & social sciences      ☐ Ecological, evolutionary & environmental sciences

For a reference copy of the document with all sections, see nature.com/documents/nr-reporting-summary-flat.pdf

# Life sciences study design

All studies must disclose on these points even when the disclosure is negative.

| | |
|---|---|
| Sample size | No sample size calculation was performed as we reanalyzed publicly available datasets and the chosen datasets consisted of hundreds of cells. The number of datasets for the downsampling and simulation benchmark was limited by the availability of appropriate tools / datasets. For the consistency benchmark, a small pilot benchmark confirmed that 10 datasets were sufficient to differentiate the performance of the transformations. |
| Data exclusions | No specific data was excluded. We applied basic quality control thresholds to the single-cell data to filter out dead cells and droplets that did not contain cells, as reported in the reproducible code. |
| Replication | All results can be reproduced using the code in https://github.com/const-ae/transformGamPoi-Paper |
| Randomization | Randomization was not relevant for our study because the computational nature of our study means that we did not perform an intervention |

| Randomization | so randomization is per definition not applicable. |
|---|---|
| Blinding | Blinding was not relevant for our study because the computational nature of our study means that we did not perform an intervention so blinding is per definition not applicable. |

# Reporting for specific materials, systems and methods

We require information from authors about some types of materials, experimental systems and methods used in many studies. Here, indicate whether each material, system or method listed is relevant to your study. If you are not sure if a list item applies to your research, read the appropriate section before selecting a response.

## Materials & experimental systems

| n/a | Involved in the study |
|---|---|
| ☒ | ☐ Antibodies |
| ☒ | ☐ Eukaryotic cell lines |
| ☒ | ☐ Palaeontology and archaeology |
| ☒ | ☐ Animals and other organisms |
| ☒ | ☐ Clinical data |
| ☒ | ☐ Dual use research of concern |

## Methods

| n/a | Involved in the study |
|---|---|
| ☒ | ☐ ChIP-seq |
| ☒ | ☐ Flow cytometry |
| ☒ | ☐ MRI-based neuroimaging |

