## [Peer review file. · Nature Methods]

Peer Review Information

Manuscript Title: Comparison of Transformations for Single-Cell RNA-Seq Data

Corresponding author name(s): Wolfgang Huber

Editorial Notes:

Reviewer Comments & Decisions:

Decision Letter, initial version:

Dear Wolfgang,

Your Analysis entitled "Transformation and Preprocessing of Single-Cell RNA-Seq Data" has now been seen by 3 reviewers, whose comments are attached. While they find your work of potential interest, they have raised serious concerns which in our view are sufficiently important that they preclude publication of the work in Nature Methods, at least in its present form.

As you will see, the reviewers agree that this is an important and timely topic but share concerns about the limited benchmarking. They also raise concerns about how the transformation methods affect downstream clustering and DE analysis.

Should further experimental data allow you to fully address these criticisms we would be willing to look at a revised manuscript (unless, of course, something similar has by then been accepted at Nature Methods or appeared elsewhere). This includes submission or publication of a portion of this work somewhere else. We hope you understand that until we have read the revised paper in its entirety we cannot promise that it will be sent back for peer-review.

If you are interested in revising this manuscript for submission to Nature Methods in the future, please contact me to discuss your appeal before making any revisions. Otherwise, we hope that you find the reviewers' comments helpful when preparing your paper for submission elsewhere.

Sincerely,
Lei

Lei Tang, Ph.D.
Senior Editor
Nature Methods

Reviewers' Comments:

Reviewer #1:

Remarks to the Author:

The intrinsic heteroscedasticity in RNA-seq data complicates their statistical analysis. Clustering analysis and visualisation of the data is not possible without prior variance stabilisation. For single cell RNA-seq data, cluster analysis and cell type identification is even the most common kind of analysis. Moreover the extreme sparsity of scRNA-seq data exaggerates the problem: a simple log transformation with an offset count (which may suffice for bulk RNA-seq data) might often create a bimodal distribution with a point mass on the offset and the rest.

In this paper, the authors describe and test different variance stabilisation approaches, that fall into 3 general categories: log + offset, regression residuals and a Bayesian inference approach. They then evaluate their performance with respect to marker gene expression distributions and the recall of K-nearest neighbour cells for simulated and downsampled real data. The authors provide a detailed discussion about the limitations of using classical performance statistics such as ROC curves and decide to focus on the recall of the 11 nearest neighbours.

In the end, the authors do not come to any clear conclusions or recommendations and finish simply by saying that all approaches have their limitations and that they expect better tools soon. All in all, the presented study is more of a review (a very well written review) than an original research paper. To make this an Analysis paper, the authors would need to extend the benchmarking part a lot.

1. The benchmark should evaluate more tangible statistics for the bench scientist, for example how would the differences in the KNN statistics be reflected clustering of cells and cell type identification and provide a more comprehensive evaluation of marker gene detection analysis.
2. It seems that the performance rankings of the methods depend on the type of data. They vary between the simulated data and the real datasets that were analysed. The authors should at least try to infer which data properties lead to those differences. What influence have library preparation methods, ie the the read or UMI count distributions that are typical for each of them? How do the results depend on the cell type composition and complexity?
3. Finally, the impact would increase a lot, if the authors would include 10X data — which is becoming the most commonly used library preparation method.

Reviewer #2:

Remarks to the Author:

The paper by Ahlmann-Eltze & Huber compares and benchmarks several different approaches to normalization of scRNA-seq UMI data: (1) the "standard" approach of library size normalization and log-transformation; (2) the recently suggested approach based on Pearson residuals; and (3) the even more recently suggested Bayesian procedure called Sanity. The authors provide an overview of strength/weaknesses of each approach and compare their performance in several benchmarks.

This topic is important and very timely: several competing normalization approaches have been recently suggested but no independent comprehensive benchmark has been carried out so far. The paper is also very clearly written and is easy to follow.

My biggest problem with the paper is that the benchmark is not comprehensive enough (see below). I think this paper could *potentially* become a very solid Nature Methods contribution in the Analysis section, if (and only if!) the benchmarking effort is substantially extended. This would require a pretty major revision. Without such extension, the paper I think is still nice and useful as an overview/primer, but may not constitute sufficient advance to appear in Nature Methods.

MAJOR ISSUES

* If the focus of the paper is on benchmarking, then the current benchmarks are not at all sufficient. The first version of the preprint did not contain Figure 4, so it seems to me that the authors originally wrote more of a "primer" and then later decided to extend it with a benchmark. That's great, but they need to do more work. In particular, there is only one measure used: kNN recall. I'd like to see more different measures.

For example, Lause et al. 2021 do a small benchmark and argue that Pearson residuals are particularly good for detecting/emphasizing rare cell types. They use a mixture dataset and use classification performance as performance metric. In fact, they use F1 macro score that specifically focuses on the rare cell type, but one could also use overall accuracy. These are metrics that the authors could use here too. They can either create other synthetic mixture datasets, or perhaps use some data with cell type labels as is.

They could in principle also insert a rare cell type directly into the kNN recall benchmark, and then compute the recall only for that rare cell type, to see if that's indeed where Pearson residuals shine (in comparison to the delta method).

Perhaps there are other possible benchmarks as well. I encourage the authors to think about it and to extend the benchmarking so that it is more comprehensive than the current Figure 4. Perhaps some benchmarks can look at the set of 1000 most variable genes selected by various methods, and assess how biologically meaningful the selected genes are. Are known markers of rare cell types selected? Etc. Would be very interesting to see this for Sanity, as Lause et al. did not include Sanity into their benchmark.

* The paper does not have a Methods section. In several places the exact processing choices were not sufficiently clear, so I think the Methods section may be needed, especially if benchmarks are extended.

* Line ~400: the authors describe what makes the Fibroblasts-1 dataset "special" but do not explain why would this give an advantage to Pearson residuals. This may be worth looking into. Is the conclusion that this is a bad dataset and the reader should rather ignore it? Or is the conclusion that Pearson residuals are very good as they can deal with this dataset?

MINOR ISSUES

* The abstract could be more specific about what was done and what is shown. What kind of benchmarks were run, on what kind datasets, which methods won, etc. "performs surprisingly well" is not specific enough: surprisingly well, but still worse than others? but better than others? etc.

* Introduction (and maybe abstract too) should be crystal clear that this paper is about UMI data normalization, not read counts like Smart-Seq. I think the word "UMI" should appear on page 1.

* I am missing a paragraph about what's the aim of THIS study in the end of the Introduction.

* Should there be a "Results" section after the "Introduction"?

* If the main selling point and contribution of the paper is benchmarking (see above) then you may want to formulate the title accordingly.

* Equation 1 -- either give a reference where this exact equation appears or (better) provide derivation somewhere, e.g. in the Appendix.

- * Box on page 2, last equation -- should it be $g'(\mu)$ not $g'(x)$? Currently there is x on the left-hand side but μ on the right-hand side.
- * Page 3, left column: maybe better connect line 86 to line 114. Already on line ~86 you could say that the simplest approach is to take $\sum y_i$ as the size factor.
- * After Equation 5: explicitly define s_j and explain that it is not estimated but simply taken as $\sum y_i$
- * Figure 2: define in the caption what exactly $\$s\$$ is and also what α as used.
- * Page 4, bullet list: regarding the 2nd (and also 3rd) bullet, Lause et al. 2021 argued that per-gene dispersion estimates are biased, and remain biased after kernel smoothing. I think you should comment here on whether this is an issue or not, as your 2nd bullet endorses approach of Hafemeister and Satija.
- * Related: what about simply using $\alpha=.1$ (or possibly estimating one value of α using the entire dataset or a subset of highly express genes)? This could be a compromise between Lause et al. (advocating using one value of α for all genes) and Hafemeister & Satija (advocating fitting alphas and not using an a priori value).
- * Figure 3: why did you use mismatched overdispersion? This needs a motivation.
- * Line 236-237: this is a potentially fair concern, however none of these things is later benchmarked in this paper! And in the benchmark on real data (Figure 4) your quantile residuals perform always worse than Pearson residuals. Do you agree?
- * Line 230 and below: it's not entirely clear why the variance of the "red" group in Figure 2 should be equal to the variance of the gray group. Maybe those red groups really do have higher variance?
- * Figure 4: what α value was used? For Pearson residuals, and also for delta methods?
- * Discussion, line 479: not accurate. Lause et al. did not use kNN "identification" (this sounds like recall), they used kNN classification accuracy. Also this paragraph should mention that Lause et al. actually focused on the F1 macro score, and there Pearson residuals performed much better.
- * References: check if any of the preprints have been published.
- * Suppl Figure S3, Sanity MAP panel. Most of the values are around zero, is that correct? Why are they not around 1? Same for Sanity distance?

* Suppl Figure S6: hardware used? libraries (for kNN search)? Again: consider adding a proper Methods section.

Reviewer #3:

Remarks to the Author:

In “Transformation and Preprocessing of Single-Cell RNA-Seq data”, the authors investigate common strategies to transform single cell count data to account for heteroskedasticity. The paper provides a clear overview of the various perspectives and functional groups of established methods as well as recently developed methods. The authors give theoretical evidence and perform various comparisons to show advantages and disadvantages of each group of methods. They conclude with general recommendations and provide code and a R package to reproduce and use their findings.

We (group leader and PhD student) have a number of comments on this interesting and timely piece of work.

Major comments:

Exposition-wise, we didn't get the point of the 3 bullets on page 4 .. is this just a list of possibilities? Seems like a strange break of text, if this could just simply be written into 3 sentences.

For us, the sequence of introducing things is very pedagogical, which is nice at times, but is at other times a bit hard to follow. For example, Figure 2 mentions methods that are not already discussed in the text (e.g., quantile residuals + SANITY are not introduced by this point). And also, there is a long section about Pearson residuals, then `sctransform`, but then the interesting 'new' part is the quantile residuals, which only gets a very small mention. The whole paper is a mix of intro / results / discussion throughout every section, as opposed to the typical flow of intro / results / discussion in separate sections.

We also struggled with the motivation of 'analyzing heteroscedastic data', because they mention 'generic statistical tests', 'least sum of squares regression' and neither of those are typically used on scRNA-seq counts. The third motivation is classification / clustering, but the authors do not actually directly attempt to address that problem (so it's good motivation, and gives the impression that it might be analyzed, but it's not).

Figure 1a seems very general. There was a connection to the logFC with increasing mean and variance in the preprint, but was removed in this version. Can a connection be made here to a downstream task/application? We think that it could help to spell out in the introduction the downstream analyses

that *are* of interest. Is it finding marker (or highly variable) genes? Or clustering? Or representative low-dimensional projections (e.g., as input to trajectory inference)? Or, all of these?

The author's main 'result' is with respect to KNN recall (including some new comparison using PCA spaces), which maybe covers many important aspects, but I have the sense that there are other important aspects / outcomes that are not directly analyzed (e.g., variance of PC1 explained by library size, clustering, feature selection performance). In the Results and Discussion, the authors mention the aspect of highly variable genes and marker gene detection and provide relevant practical and theoretical evidence for differences between different methods. Why not look into this directly / quantitatively within the benchmark?

From an overall perspective, there are also multiple dimensions to knowing whether preprocessing methods are performing well to preprocess scRNA-seq; what we are missing is some kind of a summary figure with all methods, all metrics (and all datasets).

In the discussion, the authors state "Pearson residuals are the best approach for selecting biologically meaningful genes"; at the same time, they show in Figure 2 that pearson residuals fail to stabilize the variance of genes with highly different expression between cell subtypes (marker genes). Maybe this does not affect the selection of highly variable genes, but differences in marker gene expression are used to define cell subpopulations. So, transformation methods should balance between these aspects. From the paper, it looks like randomized residuals might be able to do so, but it is not shown/investigated.

The authors mention an 'important drawback' with respect to marker genes. Can this drawback be quantified, because it's difficult to assess by just looking at Fig. 2. For example, what seems important is the ability to separate marker genes (so, DE). Perhaps a rank-based statistical test (given the different scales) could be used to quantify how well normalization methods perform for a set of known marker genes. On the other hand, it's hard also to conclude from looking at only 3 genes.

The authors state 'pearson residuals successfully rescale the data .. but heteroskedasticity remains .. may obstruct tasks like clustering, mixture modeling, DE analysis'; as mentioned above, could the authors show directly how methods affect clustering and/or DE analysis?

In Figure S1, the authors estimate the overdispersion from a set of different datasets. To limit the effect of the sequencing coverage, they only included cells between the median and 1.3x the median of the size factors. This seems to be a bit arbitrary. Wouldn't it be more realistic to divide counts by the cell size factors instead of limiting this?

'variance per gene .. is practically 0 if mean expression is less than 0.1' - Here, we wondered whether it's even important to have a stabilized variance for such lowly expressed genes? These are likely candidates to be filtered out anyways.

We have not seen log-transformed CPMs. What is the reasoning / advantage of it? Maybe add a reference?

We are not sure about the discussion around the Lause et al. simplification of the Pearson residuals. On the one hand, we find this very interesting reading and it helps to unravel the impact of single parameters within this model. On the other hand, it seems to have an excessive amount of detail compared to the randomized quantile residuals, which represents a new method and conceptual differences would be more interesting to me.

Figure 4 seems heavily focused on Smart-seq3 and scSCRB-seq data .. is there any indication that these results hold for the more prominent droplet-based (e.g., 10x) datasets? Specifically, does down-sampled Smart-seq3 structurally represent typical droplet scRNA-seq data?

Related to above, how well does the simulated data (Fig. 3) resemble real data? If we understand correctly, the authors used the simulation from the SANITY paper. But also there, I don't think the simulated data were shown to exhibit properties of real scRNA-seq. It would add credibility to show that the simulated data mimics at least some reasonable properties of real data.

Randomized Quantile residuals seem to consistently perform worse (at least compared to Pearson residuals) in the real world data benchmark, but are superior in the simulated data? Is there a good explanation for this?

Minor comments:

We found the reference to the Townes GLM-PCA method quite strange. Shouldn't it be the "Feature selection and dimension reduction for single-cell RNA-Seq based on a multinomial model" paper, which used the approach directly on scRNA-seq data (<https://genomebiology.biomedcentral.com/articles/10.1186/s13059-019-1861-6>), and not the general approach (Generalized principal component analysis; arxiv paper). Or even both?

The authors state "previous benchmarks .. focused on clustering, one of the simplest type of structure (Germain et al., 2020)". But, the Germain benchmark actually computes metrics for various other aspects and is not focused on clustering. This seems like an unfair summary.

Point-by-point response

Constantin Ahlmann-Eltze and Wolfgang Huber

Overall response

We thank all three reviewers and the editor for their time and effort they spent on our manuscript, and for their insightful comments. These have helped us to substantially improve and extend the paper. All reviewers commented positively on topic, ambition, and overall conclusions, but asked for additional, more extensive benchmarks. They also encouraged us to reformat the paper to more strictly follow the conventional structure of a research paper: Introduction, Results, Discussion, Methods. Thirdly, they encouraged us to consider applications of transformations more widely, as input for tasks including clustering, differential expression, and marker gene selection. We agree with all these points and have taken them on board in the revised version.

We have completely rewritten the paper to address the comments of the reviewers. We have significantly expanded the benchmarking section and now compare 22 transformations (representing all four major transformation classes and covering a broad range of implementations and parameter settings) using a range of performance metrics. We collected over 58,000 data-points (which took 24 years of CPU time to compute) to analyze which transformations work well and under which circumstances individual methods fail. The conclusion is clear, surprising, and we think of interest to readers: the simplest transformation, $\log(y/s+1)$, followed by a simple dimension reduction (PCA) with a suitable number of dimensions, performs as well or better than any of the more recently suggested and statistically much more sophisticated transformations.

As for the performance metrics, we consider clustering accuracy as measured by (i) adjusted Rand index, (ii) adjusted mutual information, (iii) overlap of the k nearest neighbors with a reference structure ('ground truth').

We have also carefully explored reviewers' suggestions for further performance metrics: using outcomes of differential gene expression analysis and/or of marker gene selection. For differential expression analysis, state of the art (e.g., Crowell et al. 2020, <https://doi.org/10.1038/s41467-020-19894-4>) is using edgeR or DESeq2 on the aggregated counts of 'pseudobulk' samples, which outperforms alternative approaches such as cell-level mixed models, and which works directly on the count data. Therefore, the question of transformation is of low relevance in this field, and we felt that constructing a performance metric nonetheless would be rather artificial.

A related task is the selection of marker genes, which looks for individual genes with clearly separated (e.g., bimodal) expression patterns between biologically defined cell types. However, we found it currently infeasible for us to raise this beyond anecdotal, qualitative statements into a formal metric that can be quantified objectively on multiple datasets. Complications include the biological definition of cell type and the low sampling depth of single-cell RNA-Seq, which means that even mRNAs certain to be present in a given single cell are not always sequenced and counted.

We have added an explanation of our performance metrics to the paper's Discussion section.

In the following, we provide a point-by-point response to the additional points raised by the reviewers.

Reviewer 1

Remarks to the Author: The intrinsic heteroscedasticity in RNA-seq data complicates their statistical analysis. Clustering analysis and visualisation of the data is not possible without prior variance stabilisation. For single cell RNA-seq data, cluster analysis and cell type identification is even the most common kind of analysis. Moreover the extreme sparsity of scRNA-seq data exaggerates the problem: a simple log transformation with an offset count (which may suffice for bulk RNA-seq data) might often create a bimodal distribution with a point mass on the offset and the rest.

In this paper, the authors describe and test different variance stabilisation approaches, that fall into 3 general categories: log + offset, regression residuals and a Bayesian inference approach. They then evaluate their performance with respect to marker gene expression distributions and the recall of K-nearest neighbour cells for simulated and downsampled real data. The authors provide a detailed discussion about the limitations of using classical performance statistics such as ROC curves and decide to focus on the recall of the 11 nearest neighbours.

In the end, the authors do not come to any clear conclusions or recommendations and finish simply by saying that all approaches have their limitations and that they expect better tools soon. All in all, the presented study is more of a review (a very well written review) than an original research paper. To make this an Analysis paper, the authors would need to extend the benchmarking part a lot.

We want to thank the reviewer for their comments. We have expanded the benchmarking and put it at the center of the paper. We are now also stating a much more pointed conclusion, namely, that one of the simplest methods works as well as or better than more complicated ones. We believe this conclusion is non-obvious, may be unexpected, and will be broadly useful to readers of the journal.

1. The benchmark should evaluate more tangible statistics for the bench scientist, for example how would the differences in the KNN statistics be reflected clustering of cells and cell type identification and provide a more comprehensive evaluation of marker gene detection analysis.

We have addressed these suggestions as outlined above in the general part of this response.

2. It seems that the performance rankings of the methods depend on the type of data. They vary between the simulated data and the real datasets that were analysed. The authors should at least try to infer which data properties lead to those differences. What influence have library preparation methods, ie the the read or UMI count distributions that are typical for each of them? How do the results depend on the cell type composition and complexity?

With the expanded datasets, the results of the simulations and the data-based benchmarks agree well. The supposed disagreements in the earlier version were essentially measurement fluctuations, which highlights the importance of using a varied set of data for benchmarking.

We thank the reviewer for suggesting to check what influence the library size has on the performance of the transformation. In Figure 4D, we show the performance of different transformations stratified by sequencing depth per cell, and Suppl. Figs. S10–S12 show the performances depending on the cell type complexity.

3. Finally, the impact would increase a lot, if the authors would include 10X data — which is becoming the most commonly used library preparation method.

We have included a new type of benchmark on 10X datasets. Instead of measuring the performance with respect to ground truth (which is generally not available for this type of data), we used a ‘data splitting’ approach on ten diverse datasets. For each of them, we measured the consistency of the k nearest neighbor graph after transformation on two distinct data halves, by splitting the genes into two disjoint subsets. While consistency is not *sufficient* for good performance, it is a *necessary* property of a good method, and thus performance on this measure provides an upper limit for method quality and is a helpful metric for method comparison. Moreover, we have rewritten the section on the downsampling

benchmark (performed on datasets produced with experimental protocols that provide higher per-cell coverage and that differ from standard 10X protocols) to emphasize its relevance for 10X data.

Reviewer 2

Remarks to the Author: The paper by Ahlmann-Eltze & Huber compares and benchmarks several different approaches to normalization of scRNA-seq UMI data: (1) the “standard” approach of library size normalization and log-transformation; (2) the recently suggested approach based on Pearson residuals; and (3) the even more recently suggested Bayesian procedure called Sanity. The authors provide an overview of strength/weaknesses of each approach and compare their performance in several benchmarks.

This topic is important and very timely: several competing normalization approaches have been recently suggested but no independent comprehensive benchmark has been carried out so far. The paper is also very clearly written and is easy to follow.

My biggest problem with the paper is that the benchmark is not comprehensive enough (see below). I think this paper could *potentially* become a very solid Nature Methods contribution in the Analysis section, if (and only if!) the benchmarking effort is substantially extended. This would require a pretty major revision. Without such extension, the paper I think is still nice and useful as an overview/primer, but may not constitute sufficient advance to appear in Nature Methods.

MAJOR ISSUES

- If the focus of the paper is on benchmarking, then the current benchmarks are not at all sufficient. The first version of the preprint did not contain Figure 4, so it seems to me that the authors originally wrote more of a “primer” and then later decided to extend it with a benchmark. That’s great, but they need to do more work. In particular, there is only one measure used: kNN recall. I’d like to see more different measures.

We have followed the reviewer’s suggestion for a major revision of the paper. We have condensed the “primer” part about the transformations into a more traditional introduction section and have significantly expanded the benchmark section, as outlined above in the general part of this response.

For example, Lause et al. 2021 do a small benchmark and argue that Pearson residuals are particularly good for detecting/emphasizing rare cell types. They use a mixture dataset and use classification performance as performance metric. In fact, they use F1 macro score that specifically focuses on the rare cell type, but one could also use overall accuracy. These are metrics that the authors could use here too. They can either create other synthetic mixture datasets, or perhaps use some data with cell type labels as is.

We thank the reviewer for these exemplary suggestions. We have studied them carefully and concluded to proceed as follows to achieve the common goal, i.e., additional performance metrics:

The paper by Lause et al. (2021) is excellent, also their benchmark approach is interesting and serves the purpose of supporting their claims. However, we were concerned about adapting it to our work for two reasons:

- The metric relies on FACS annotation of cell types as ground truth; however, it is not guaranteed that FACS-based gating is necessarily more correct (biologically relevant) than transcriptome-based annotation. This is, in fact, a generic concern with using “ground truth” cell types derived from other, older technologies, and not a specific critique of the Lause et al. (2021) paper.
- The synthetic “rare” cell type is created by copying the count data of 50 existing B-cells and adding 10 genes exclusively expressed in the new cell type. Here, however, we did not want to rely on the assumption that such a localized, pointed change mimics biologically relevant cell type/state differences, due to the high interconnectedness of biological processes in the cell and the nature of gene expression regulation networks. Again, this is a generic concern: although the idea is appealing, we think that our understanding of biology is not yet at a place where we could confidently take the sc-RNA-Seq data from a given cell type (or state) and computationally turn it into another.

Given these considerations, we decided to

1. reuse established (in 4 of 5 cases: previously published) simulation frameworks, and
2. use deeply sequenced data as a reference and downsample them to mimic 10X-like data, which is the major data type of interest for the transformation-based approaches that our manuscript is about.

They could in principle also insert a rare cell type directly into the kNN recall benchmark, and then compute the recall only for that rare cell type, to see if that's indeed where Pearson residuals shine (in comparison to the delta method).

We thank the reviewer for this excellent suggestion. We took it up and looked at the performance of each transformation as a function of cluster size. Suppl. Figs. S10-S12 show the results for k-NN overlap on three datasets, one for each benchmark type. While we do see interesting variations in performance, we did not find that a single transformation always performed particularly well or badly, a finding that is somewhat in contrast to that of Lause et al.

Perhaps there are other possible benchmarks as well. I encourage the authors to think about it and to extend the benchmarking so that it is more comprehensive than the current Figure 4. Perhaps some benchmarks can look at the set of 1000 most variable genes selected by various methods, and assess how biologically meaningful the selected genes are. Are known markers of rare cell types selected? Etc. Would be very interesting to see this for Sanity, as Lause et al. did not include Sanity into their benchmark.

In our expanded benchmarks, we consider three variations of the standard Pearson residual and shifted logarithm transformation. We combine them with highly variable gene selection, z-scoring, and z-scoring after highly variable gene selection. We find that none of the post-processing approaches improves k-NN overlap; the highly variable gene selection only makes calculating the PCA a bit faster.

We also explored the selection of known marker genes as a benchmark criterion, but found it too subjective and manipulatable (“cherry-picking”), and did not further pursue it, as outlined above in the general part of this response.

- The paper does not have a Methods section. In several places the exact processing choices were not sufficiently clear, so I think the Methods section may be needed, especially if benchmarks are extended.

We have now included a detailed method section at the end of the manuscript.

- Line ~400: the authors describe what makes the Fibroblasts-1 dataset “special” but do not explain why would this give an advantage to Pearson residuals. This may be worth looking into. Is the conclusion that this is a bad dataset and the reader should rather ignore it? Or is the conclusion that Pearson residuals are very good as they can deal with this dataset?

In our extended benchmark, we have included an additional Smart-seq3 dataset that contains many more cells and confirmed our previous results. Instead of the individual results, we now present in the main text the aggregated data, which on average, show similar performance for delta method-based transformations and the Pearson residuals.

The Fibroblast dataset from Hagemann-Jensen et al. (2020) is, in some sense, special: it has the simplest latent structure (Suppl. Fig. S6) and nonetheless shows the highest overdispersion (Suppl. Tab. S3). In the meanwhile, we suspect that it is a low-quality dataset, but given that we have started using it, we did not want to evoke the impression of selection bias by now excluding it. However, we also do not want to overemphasize its importance and prefer not to give it extra attention, as we do not expect that conclusions specifically drawn from this dataset are of wider interest.

MINOR ISSUES

- The abstract could be more specific about what was done and what is shown. What kind of benchmarks were run, on what kind datasets, which methods won, etc. “performs surprisingly well” is not specific enough: surprisingly well, but still worse than others? but better than others? etc.

We have rephrased the abstract. In particular, we now explicitly report that the shifted logarithm performs as well or better than the alternative approaches.

- Introduction (and maybe abstract too) should be crystal clear that this paper is about UMI data normalization, not read counts like Smart-Seq. I think the word “UMI” should appear on page 1.

We have updated the Introduction to clarify that we compare transformations for UMI data.

- I am missing a paragraph about what’s the aim of THIS study in the end of the Introduction.

We have added a paragraph at the end of the Introduction explaining what this study is about.

- Should there be a “Results” section after the “Introduction”?

Yes. We have completely rewritten the paper, which now follows the classical Introduction-Results-Discussion structure.

- If the main selling point and contribution of the paper is benchmarking (see above) then you may want to formulate the title accordingly.

We agree and have changed the title to “Comparison of Transformations for Single-Cell RNA-Seq Data”, which emphasizes the comparative nature of our paper.

- Equation 1 – either give a reference where this exact equation appears or (better) provide derivation somewhere, e.g. in the Appendix.

We have added a section in the Appendix that explains how this function is derived from a quadratic mean-variance relation, and are referring to it where Equation (1) is presented.

- Box on page 2, last equation – should it be $g'(\mu)$ not $g'(x)$? Currently there is x on the left-hand side but μ on the right-hand side.

Yes. We apologize for this typographical error. We have fixed it so that the left-hand side now uses $g'(\mu)$.

- Page 3, left column: maybe better connect line 86 to line 114. Already on line ~86 you could say that the simplest approach is to take $\sum y_i$ as the size factor.

We have moved the explanation of the counts-per-million into the Introduction.

- After Equation 5: explicitly define s_j and explain that it is not estimated but simply taken as $\sum y_i$.

The size factor estimator we used throughout this work is now defined in Equation (3) in the Introduction section, and it is essentially the sum of counts per cell. We are well aware of alternative estimators, e.g., Lun, Bach and Marioni (<https://doi.org/10.1186/s13059-016-0947-7>) or <http://bioconductor.org/books/3.15/OSCA.basic/normalization.html#library-size-normalization>. The different approaches may lead to markedly different results for some “extreme” cells. However, for the majority of datasets we are considering, such differences will be small and/or rare, and not lead to changes in the relative ranking of our aggregate performance metrics. Thus, to not further expand the complexity of the benchmarking, we have fixed it to this widely used, and arguably simplest and most obvious, choice.

- Figure 2: define in the caption what exactly s is and also what α as used.

We now detail in the Methods section which parameters we used for the transformations presented in the figures.

- Page 4, bullet list: regarding the 2nd (and also 3rd) bullet, Lause et al. 2021 argued that per-gene dispersion estimates are biased, and remain biased after kernel smoothing. I think you should comment here on whether this is an issue or not, as your 2nd bullet endorses approach of Hafemeister and Satija.

We have replaced the previous discussion on setting the overdispersion with a quantitative analysis that compares the performance of fixing the overdispersion to zero or some other value.

- Related: what about simply using $\alpha=0.1$ (or possibly estimating one value of α using the entire dataset or a subset of highly express genes)? This could be a compromise between Lause et al. (advocating using one value of alpha for all genes) and Hafemeister & Satija (advocating fitting alphas and not using an a priori value).

In our benchmark, we now include setting $\alpha=0.05$ and find that it performs better than fixing $\alpha = 0$ and performs similar to estimating the overdispersion from the data.

- Figure 3: why did you use mismatched overdispersion? This needs a motivation.

We did not want to bias the results in favor of the methods with a fixed overdispersion. We have added a brief statement in the Methods section explaining why we chose the specific overdispersion for the Random Walk and Linear Walk benchmarks.

- Line 236-237: this is a potentially fair concern, however none of these things is later benchmarked in this paper! And in the benchmark on real data (Figure 4) your quantile residuals perform always worse than Pearson residuals. Do you agree?

We agree with the reviewer. The new manuscript retains a qualitative/conceptual note on the linear nature of the Pearson residuals-based transformations, but this is not something that appears to affect the benchmarks. And yes, the randomized quantile residuals perform consistently worse than Pearson residuals.

- Line 230 and below: it's not entirely clear why the variance of the "red" group in Figure 2 should be equal to the variance of the gray group. Maybe those red groups really *do* have higher variance?

Yes - this is the point. If two groups of cells have different mean expression levels for a particular gene, i.e., if there is a bimodal expression pattern such as in what is now Figure 1C, and if monotonous non-linear transformations of the data are admissible (like the logarithm), then the relation between the gene's variances in the two groups can be either way. On the "raw" (counts) scale, the group with the higher mean will generally have higher variance, but on a transformed scale, this relationship may be the other way round, or the variances may be equal. For modeling purposes, roughly equal variances can be advantageous, and that is generally used as an argument for transformations such as the logarithm. The point of our discussion here is that the Pearson residuals, since they are affine linear, do not offer this flexibility. Fig. 1C is intended to exemplify this on a real data example. We can also provide additional illustration using the following small simulation.

```
library("dplyr"); library("ggplot2")
d = tibble(group = rep(c("A", "B"), each = 1000),
           y = rpois(n = length(group), lambda = c(A = 5, B = 50)[group]),
           what = "orig")
rbind(d,
      mutate(d, y = (y - mean(y)) / sd(y), what = "Pearson"),
      mutate(d, y = sqrt(y), what = "sqrt (delta method)")) |>
ggplot(aes(x = y, fill = group)) +
  geom_histogram(bins = 32) +
  facet_wrap(vars(what), ncol = 3, scales = "free")
```

- Figure 4: what alpha value was used? For Pearson residuals, and also for delta methods?

We added a paragraph to the Methods section explaining what overdispersion (alpha) we used.

- Discussion, line 479: not accurate. Lause et al. did not use kNN “identification” (this sounds like recall), they used kNN classification accuracy. Also this paragraph should mention that Lause et al. actually focused on the F1 macro score, and there Pearson residuals performed much better.

We thank the reviewer for spotting the inaccuracy. Indeed, Lause et al. did not use k-NN identification, but measured the classification accuracy using overall accuracy and a macro F1 score. We have added a paragraph in the discussion that explains how the results of Lause et al. relate to our benchmark results.

- References: check if any of the preprints have been published.

We have checked our references and ensured that we cite the published versions of papers.

- Suppl Figure S3, Sanity MAP panel. Most of the values are around zero, is that correct? Why are they not around 1? Same for Sanity distance?

The variance for the Sanity MAP in what is now Suppl. Fig. S2 depends on how much Sanity’s estimates of the log-fold changes (δ_{gc}) vary for a gene. The estimates are regularized using a prior (see Section S1.1.2 of Breda et al.), which can result in their variances being near zero. Suppl. Fig. S2 also shows the pattern for Sanity Distance. Here, the variances are larger because accounting for the uncertainty of δ_{gc} , which is large for lowly expressed genes, increases the gene-wise variance.

- Suppl Figure S6: hardware used? libraries (for kNN search)? Again: consider adding a proper Methods section.

We have added a complete method section detailing the hardware and the specific software packages we used.

Reviewer 3

Remarks to the Author: In “Transformation and Preprocessing of Single-Cell RNA-Seq data”, the authors investigate common strategies to transform single cell count data to account for heteroskedasticity. The paper provides a clear overview of the various perspectives and functional groups of established methods as well as recently developed methods. The authors give theoretical evidence and perform various comparisons to show advantages and disadvantages of each group of methods. They conclude with general recommendations and provide code and a R package to reproduce and use their findings.

We (group leader and PhD student) have a number of comments on this interesting and timely piece of work.

Major comments:

Exposition-wise, we didn't get the point of the 3 bullets on page 4 .. is this just a list of possibilities? Seems like a strange break of text, if this could just simply be written into 3 sentences.

For us, the sequence of introducing things is very pedagogical, which is nice at times, but is at other times a bit hard to follow. For example, Figure 2 mentions methods that are not already discussed in the text (e.g., quantile residuals + SANITY are not introduced by this point). And also, there is a long section about Pearson residuals, then `sctransform`, but then the interesting 'new' part is the quantile residuals, which only gets a very small mention. The whole paper is a mix of intro / results / discussion throughout every section, as opposed to the typical flow of intro / results / discussion in separate sections.

We thank Reviewer Team 3 for these thoughtful comments. We have completely overhauled the manuscript, which now follows a more familiar structure.

We also struggled with the motivation of ‘analyzing heteroscedastic data’, because they mention ‘generic statistical tests’, ‘least sum of squares regression’ and neither of those are typically used on scRNA-seq counts. The third motivation is classification / clustering, but the authors do not actually directly attempt to address that problem (so it's good motivation, and gives the impression that it might be analyzed, but it's not).

We have rewritten this section to make the motivation clearer.

Figure 1a seems very general. There was a connection to the logFC with increasing mean and variance in the preprint, but was removed in this version. Can a connection be made here to a downstream task/application? We think that it could help to spell out in the introduction the downstream analyses that *are* of interest. Is it finding marker (or highly variable) genes? Or clustering? Or representative low-dimensional projections (e.g., as input to trajectory inference)? Or, all of these?

We have removed this figure as it was indeed of a more review-like character and not essential for the current, more streamlined structure. We have clarified throughout the text that our benchmark is concerned with testing how evident the latent structure of the data is after each transformation, as measured by the k nearest neighbor overlap using the Euclidean metric.

The authors' main 'result' is with respect to KNN recall (including some new comparison using PCA spaces), which maybe covers many important aspects, but I have the sense that there are other important aspects / outcomes that are not directly analyzed (e.g., variance of PC1 explained by library size, clustering, feature selection performance). In the Results and Discussion, the authors mention the aspect of highly variable genes and marker gene detection and provide relevant practical and theoretical evidence for differences between different methods. Why not look into this directly / quantitatively within the benchmark?

We thank Reviewer Team 3 for these suggestions. We have significantly expanded the benchmark and now analyze many more aspects. In Fig. 1A and Suppl. Fig. S1, we look at the correlation between the principal components and the library size. In Suppl. Fig. S9, we compare the performance of the transformations for clustering, measured by the adjusted Rand index and adjusted mutual information,

of the simulated data. Figure 4D considers the performance on datasets stratified by size factor per cell. We have clarified in the paper why we do not focus on marker gene selection, as also explained above in the general part of this response letter.

From an overall perspective, there are also multiple dimensions to knowing whether preprocessing methods are performing well to preprocess scRNA-seq; what we are missing is some kind of a summary figure with all methods, all metrics (and all datasets).

In the new version, we follow this advice: Figure 1 provides an overview of the conceptual differences between the three transformation families. Figure 2 gives an overview of the results from the three benchmarks.

In the discussion, the authors state “Pearson residuals are the best approach for selecting biologically meaningful genes”; at the same time, they show in Figure 2 that Pearson residuals fail to stabilize the variance of genes with highly different expression between cell subtypes (marker genes). Maybe this does not affect the selection of highly variable genes, but differences in marker gene expression are used to define cell subpopulations. So, transformation methods should balance between these aspects. From the paper, it looks like randomized residuals might be able to do so, but it is not shown/investigated.

Indeed, the statement that the reviewers refer to was a quote: “They [Lause et al.] concluded that Pearson residuals are the best approach for selecting biologically meaningful genes” (line 473), and then we went on to discuss (and dispute) the validity of that claim. We hope that in the revised version, this is now clearer.

We have also clarified in the paper why we do not focus on marker gene selection, as also explained above in the general part of this response letter.

The authors mention an ‘important drawback’ with respect to marker genes. Can this drawback be quantified, because it’s difficult to assess by just looking at Fig. 2. For example, what seems important is the ability to separate marker genes (so, DE). Perhaps a rank-based statistical test (given the different scales) could be used to quantify how well normalization methods perform for a set of known marker genes. On the other hand, it’s hard also to conclude from looking at only 3 genes.

We have rephrased that paragraph as we cannot really quantify how big that drawback is. We now make clear it’s a conceptual and perhaps even only aesthetic concern. (Note, however, that a rank-based test corresponds to rank-transforming the data, which is yet another transformation.)

The authors state ‘pearson residuals successfully rescale the data .. but heteroskedasticity remains .. may obstruct tasks like clustering, mixture modeling, DE analysis’; as mentioned above, could the authors show directly how methods affect clustering and/or DE analysis?

We have rephrased this sentence to ensure we only make claims about things we have actually measured (i.e., k-NN overlap and clustering).

In Figure S1, the authors estimate the overdispersion from a set of different datasets. To limit the effect of the sequencing coverage, they only included cells between the median and 1.3x the median of the size factors. This seems to be a bit arbitrary. Wouldn’t it be more realistic to divide counts by the cell size factors instead of limiting this?

Indeed, this would be an alternative way to proceed here, but here we wanted to study overdispersion explicitly without the confounding effect of such division/scaling operations. We have added the reasoning behind this to the manuscript.

‘variance per gene .. is practically 0 if mean expression is less than 0.1’ - Here, we wondered whether it’s even important to have a stabilized variance for such lowly expressed genes? These are likely candidates to be filtered out anyways.

We agree that these very lowly expressed genes do not contribute much to the result. However, defining a threshold beyond which genes are to be ignored is difficult. There is always the danger of removing informative genes. As datasets get larger, it is feasible that some rarer cell types are characterized by genes with high cell-type specific but low average expression. Instead, the PCA does an excellent job of picking up consistent patterns across genes and ignoring irrelevant noise in the data, as illustrated by comparing the performance of the $\log(x + 1)$ with and without highly variable genes (HVG) selection.

We have not seen log-transformed CPMs. What is the reasoning / advantage of it? Maybe add a reference?

We were motivated by a highly-cited review by Luecken et al. (2019), titled “Current best practices in single-cell RNA-seq analysis: a tutorial”, which called the counts per million normalization “the most commonly used normalization protocol”. Although the superlative may be debatable, as, e.g., the default normalization in Seurat and Scanpy use counts per ten-thousands, the prominence of the review and further examples in the literature (Zhang et al. (2019), Muller et al. (2017), Tung et al. (2017)) each with more than a hundred citations, compelled us to include the counts per million in our benchmark.

We are not sure about the discussion around the Lause et al. simplification of the Pearson residuals. On the one hand, we find this very interesting reading and it helps to unravel the impact of single parameters within this model. On the other hand, it seems to have an excessive amount of detail compared to the randomized quantile residuals, which represents a new method and conceptual differences would be more interesting to me.

We thank the reviewer for this comment and fully agree. Accordingly, we have shortened the discussion of the debate between Lause et al. and Hafemeister et al.

Figure 4 seems heavily focused on Smart-seq3 and scSCRB-seq data .. is there any indication that these results hold for the more prominent droplet-based (e.g., 10X) datasets? Specifically, does down-sampled Smart-seq3 structurally represent typical droplet scRNA-seq data?

We have included a new type of benchmark in our paper that uses 10X data. Furthermore, our Suppl. Figs. S5 and S6, and Suppl. Table S3 show that many summary statistics of the downsampled Smart-seq3 data match typical droplet scRNA-seq data.

Related to above, how well does the simulated data (Fig. 3) resemble real data? If we understand correctly, the authors used the simulation from the SANITY paper. But also there, I don’t think the simulated data were shown to exhibit properties of real scRNA-seq. It would add credibility to show that the simulated data mimics at least some reasonable properties of real data.

We have now included three more benchmarking frameworks to increase the diversity of our simulated datasets. Suppl. Fig. S5 and Suppl. Table S3 show that many summary statistics of the simulated data match real data—however, Suppl. Fig. S6 shows that the latent structure of the simulated data often differed from real datasets (either being too simple (Dyngen, muscat) or too complex (Linear Walk, Random Walk). Yet, as the goal of our paper was not to compare simulation frameworks, we have decided not to further elaborate on this point in the manuscript.

Randomized Quantile residuals seem to consistently perform worse (at least compared to Pearson residuals) in the real world data benchmark, but are superior in the simulated data? Is there a good explanation for this?

In our latest round of benchmarks, the randomized quantile residuals perform similarly in the consistency benchmark but worse than Pearson residuals in the simulated and downsampling benchmark.

Minor comments:

We found the reference to the Townes GLM-PCA method quite strange. Shouldn't it be the "Feature selection and dimension reduction for single-cell RNA-Seq based on a multinomial model" paper, which used the approach directly on scRNA-seq data (<https://genomebiology.biomedcentral.com/articles/10.1186/s13059-019-1861-6>), and not the general approach (Generalized principal component analysis; arxiv paper). Or even both?

We thank the reviewers for spotting this oversight. We now cite both papers by Townes.

The authors state "previous benchmarks .. focused on clustering, one of the simplest type of structure (Germain et al., 2020)". But, the Germain benchmark actually computes metrics for various other aspects and is not focused on clustering. This seems like an unfair summary.

Indeed. We have rephrased the sentence as it was ambiguous. We have now clarified that even though Germain et al. (2020) analyzed the impact of many different steps of the processing pipeline, their benchmarks ultimately always refer back to the a priori assigned cell populations (e.g., Fig. 1B of Germain et al. (2020)).

References

Crowell et al. 2020. "muscat detects subpopulation-specific state transitions from multi-sample multi-condition single-cell transcriptomics data", Nature Communications <https://doi.org/10.1038/s41467-020-19894-4>

Hagemann-Jensen et al. 2020. "Single-cell RNA counting at allele and isoform resolution using Smart-seq3", Nature Biotechnology <https://doi.org/10.1038/s41587-020-0497-0>

Luecken et al. 2019. "Current best practices in single-cell RNA-seq analysis: a tutorial", Molecular System Biology <https://www.embopress.org/doi/full/10.15252/msb.20188746>

Zhang et al. 2019. "Defining inflammatory cell states in rheumatoid arthritis joint synovial tissues by integrating single-cell transcriptomics and mass cytometry", Nature Immunology <https://www.nature.com/articles/s41590-019-0378-1>

Muller et al. 2017. "Single-cell profiling of human gliomas reveals macrophage ontogeny as a basis for regional differences in macrophage activation in the tumor microenvironment", Genome Biology <https://link.springer.com/article/10.1186/s13059-017-1362-4>

Tung et al. 2017. "Batch effects and the effective design of single-cell gene expression studies", Scientific Reports <https://www.nature.com/articles/srep39921>

Decision Letter, first revision:

Dear Dr Huber,

I apologize for the delay in getting back to you. Thank you for your letter asking us to reconsider our decision on your Analysis, "Transformation and Preprocessing of Single-Cell RNA-Seq Data". After careful consideration we have decided that we are willing to consider a revised version of your manuscript and will send it back to the original reviewers. .

Please resubmit all the necessary files electronically by using the link below to access your home page

[REDACTED]

When submitting you paper please

- * include a point-by-point response to our referees and to any editorial suggestions
- * please underline/highlight any additions to the text or areas with other significant changes to facilitate review of the revised manuscript
- * address the points listed described below to conform to our open science requirements
- * ensure it complies with our general format requirements as set out in our guide to authors at www.nature.com/naturemethods

OPEN SCIENCE REQUIREMENTS

REPORTING SUMMARY AND EDITORIAL POLICY CHECKLISTS

When revising your manuscript, please submit reporting summary and editorial policy checklists.

Please note that these forms are dynamic 'smart pdfs' and must therefore be downloaded and

completed in Adobe Reader. We will then flatten them for ease of use by the reviewers. If you would like to reference the guidance text as you complete the template, please access these flattened versions at <http://www.nature.com/authors/policies/availability.html>.

DATA AVAILABILITY

CODE AVAILABILITY

Please include a "Code Availability" subsection in the Online Methods which details how your custom code is made available. Only in rare cases (where code is not central to the main conclusions of the paper) is the statement "available upon request" allowed (and reasons should be specified).

For more information on our code sharing policy and requirements, please see: <https://www.nature.com/nature-research/editorial-policies/reporting-standards#availability-of-computer-code>

SUPPLEMENTARY PROTOCOL

To help facilitate reproducibility and uptake of your method, we ask you to prepare a step-by-step Supplementary Protocol for the method described in this paper. We [encourage authors to share their step-by-step experimental protocols](https://www.nature.com/nature-research/editorial-policies/reporting-standards#protocols) on a protocol sharing platform of their choice and report the protocol DOI in the reference list. Nature Portfolio 's Protocol Exchange is a free-to-use and open resource for protocols; protocols deposited in Protocol Exchange are citable and can be linked from the published article. More details can found at www.nature.com/protocolexchange/about.

ORCID

Nature Methods is committed to improving transparency in authorship. As part of our efforts in this direction, we are now requesting that all authors identified as 'corresponding author' on published papers create and link their Open Researcher and Contributor Identifier (ORCID) with their account on the Manuscript Tracking System (MTS), prior to acceptance. This applies to primary research papers

only. ORCID helps the scientific community achieve unambiguous attribution of all scholarly contributions. You can create and link your ORCID from the home page of the MTS by clicking on 'Modify my Springer Nature account'. For more information please visit www.springernature.com/orcid.

Best regards,
Lei

Lei Tang, Ph.D.
Senior Editor
Nature Methods

Author Rebuttal, first revision:

[There is no response to the referee letter at this stage.]

Decision Letter, second revision:

Dear Dr Huber,

Your revised Analysis, "Comparison of Transformations for Single-Cell RNA-Seq Data", has now been seen by the original three reviewers again. As you will see from their comments below, two reviewers support publication, while reviewer #2 remains not fully convinced about the conclusions made in the manuscript. We are interested in the possibility of publishing your paper in Nature Methods, but would like to consider your response to these concerns before we reach a final decision on publication.

We therefore invite you to revise your manuscript to address these concerns raised by reviewer #2. We are committed to providing a fair and constructive peer-review process. Do not hesitate to contact me if there are specific requests from the reviewer #2 that you believe are technically impossible or unlikely to yield a meaningful outcome.

- * include a point-by-point response to the reviewers and to any editorial suggestions
- * please underline/highlight any additions to the text or areas with other significant changes to facilitate

review of the revised manuscript

- * address the points listed described below to conform to our open science requirements
- * ensure it complies with our general format requirements as set out in our guide to authors at www.nature.com/naturemethods
- * resubmit all the necessary files electronically by using the link below to access your home page

[REDACTED]

We hope to receive your revised paper within 4 weeks. If you cannot send it within this time, please let us know. In this event, we will still be happy to reconsider your paper at a later date so long as nothing similar has been accepted for publication at Nature Methods or published elsewhere.

OPEN SCIENCE REQUIREMENTS

REPORTING SUMMARY AND EDITORIAL POLICY CHECKLISTS

Please note that these forms are dynamic ‘smart pdfs’ and must therefore be downloaded and completed in Adobe Reader. We will then flatten them for ease of use by the reviewers. If you would like to reference the guidance text as you complete the template, please access these flattened versions at <http://www.nature.com/authors/policies/availability.html>.

DATA AVAILABILITY

All novel DNA and RNA sequencing data, protein sequences, genetic polymorphisms, linked genotype and phenotype data, gene expression data, macromolecular structures, and proteomics data must be deposited in a publicly accessible database, and accession codes and associated hyperlinks must be provided in the “Data Availability” section.

Please include a “Data availability” subsection in the Online Methods. This section should inform readers about the availability of the data used to support the conclusions of your study, including accession codes to public repositories, references to source data that may be published alongside the paper, unique identifiers such as URLs to data repository entries, or data set DOIs, and any other statement about data availability. At a minimum, you should include the following statement: “The data that support the findings of this study are available from the corresponding author upon request”, describing which data is available upon request and mentioning any restrictions on availability. If DOIs are provided, please include these in the Reference list (authors, title, publisher (repository name), identifier, year). For more guidance on how to write this section please see:

<http://www.nature.com/authors/policies/data/data-availability-statements-data-citations.pdf>

CODE AVAILABILITY

Please include a “Code Availability” subsection in the Online Methods which details how your custom code is made available. Only in rare cases (where code is not central to the main conclusions of the paper) is the statement “available upon request” allowed (and reasons should be specified).

SUPPLEMENTARY PROTOCOL

To help facilitate reproducibility and uptake of your method, we ask you to prepare a step-by-step Supplementary Protocol for the method described in this paper. We [encourage authors to share their step-by-step experimental protocols](https://www.nature.com/nature-research/editorial-policies/reporting-standards#protocols) on a protocol sharing platform of their choice and report the protocol DOI in the reference list. Nature Portfolio's Protocol Exchange is a free-to-use and open resource for protocols; protocols deposited in Protocol Exchange are citable and can be linked from the published article. More details can be found at www.nature.com/protocolexchange/about.

ORCID

Nature Methods is committed to improving transparency in authorship. As part of our efforts in this direction, we are now requesting that all authors identified as ‘corresponding author’ on published papers create and link their Open Researcher and Contributor Identifier (ORCID) with their account on the Manuscript Tracking System (MTS), prior to acceptance. This applies to primary research papers only. ORCID helps the scientific community achieve unambiguous attribution of all scholarly contributions. You can create and link your ORCID from the home page of the MTS by clicking on ‘Modify my Springer Nature account’. For more information please visit www.springernature.com/orcid.

Best regards,
Lei

Lei Tang, Ph.D.
Senior Editor
Nature Methods

Reviewers' Comments:

Reviewer #1:

Remarks to the Author:

The manuscript has been vastly improved, it no longer reads like a review, but includes many additional analyses on relevant data-sets. For readers with specific questions, the authors provide a shiny app so that everybody can browse the benchmarking results.

Finally, the discussion states clear recommendations for the bench scientist. All in all, the manuscript can be published as is.

Reviewer #2:

Remarks to the Author:

The revised version of the paper by Ahlmann-Eltze & Huber is a comprehensive benchmark of different approaches to normalization of scRNA-seq UMI data. As I wrote before, the topic is important and very timely. In the first round, all reviewers asked to make the benchmark more comprehensive, and the authors did a very good job at addressing this point. The paper, again, is very clearly written, easy to follow, and the figures are concise and masterful.

That said, I am not convinced that the presented evidence supports the authors' conclusions (which are, as the authors admit, nontrivial and surprising). Below I describe what I think is currently missing. Without these results, the paper, in my opinion, is inconclusive. I request some simple experiments, and I think their outcome would determine whether the authors' conclusions hold up or not.

Overall my worry is that the authors were so strict about what can be taken as "ground truth" (line 279) that they threw the baby out with the bathwater and were only left with benchmark metrics that are insufficiently sensitive. See below. But the authors may be able to convince me otherwise.

MAJOR ISSUES

1) The main conclusion of the paper is that $\log_1 p$ transform performs as well or better than the fancy alternatives. This conclusion is mainly based on Fig 2a (consistency benchmark) that shows that $\log_1 p$ gives more "consistent" results. However, it is logically possible that a method gives consistent but at the same time misleading results; this could e.g. be the case if $\log_1 p$ is strongly (but consistently) influenced by the size factor. This concern is not addressed at all.

My suggestion would be to include "negative controls" into Figure 2, namely include rows for the "raw data" ($\$y\$$) and maybe also the normalized data ($\$y/s\$$). The authors say that the UMI data need to be transformed, but we do not actually see this in Figure 2. Would $\$y\$$ and $\$y/s\$$ perform much worse than everything else? In particular, would they be less "consistent" in Fig 2a? If $\$y\$$ or $\$y/s\$$ turns out to be as consistent as the $\log_1 p$, then this would, I think, suggest that the consistency metric is not very useful.

2) Related: the authors do conclude that $\log(\text{CPM}+1)$ performs "poorly" (line 488), but what is this assessment based on? In Figure 2A, $\log\text{CPM}$ performs very well, and it is not too bad in 2B and 2C either. I certainly agree that $\log\text{CPM}$ is a bad idea for UMI data, but I can hardly see it in Figure 2, and so I worry that the benchmark may not be sensitive enough to actually distinguish good normalization methods from bad ones.

3) Figure 1A shows that $\log_1 p$ transformation does not remove the confounding effect of the size factor. Why do the authors not turn it into a formal benchmark? In Figure S1 they even have a numerical metric to quantify the confounding (canonical correlation). I think it would be very interesting to see a figure in the style of Figure 2, which would show this canonical correlation for different transformations for all analyzed datasets. This is an objective performance metric, and very likely delta method would perform poorly there. This I think deserves to be shown and then discussed.

4) This may be a less important issue, but I am surprised that the intersection of all kNN neighborhoods for Fig in 2C was non-empty! Line 1023: how many neighbors did you have in the intersection? On average? This needs to be reported, maybe show a histogram. Further, what fraction of cells did actually have more than one reliable nearest neighbor (line 1034)? If it was a very small fraction, does it not undermine the usefulness of this approach? Please report and discuss.

MINOR ISSUES

* Line 233: valid argument, but the first and the third histograms in Figure 1C are very obviously *not* a linear transformation of each other. Why? I assume it is because Pearson residuals are linear transformation only assuming that all size factors are equal to 1, while in Fig 1C they are likely not. I think this needs to be clarified in the text, and maybe also illustrated somehow (additional row in Figure 1 that would be similar to 1C but only use cells with nearly the same size factors? or maybe insert the figure from the rebuttal into the supplementary materials?)

* line 335: "four to eight dimension settings for the PCA" -- this is unclear. This reads as if you used up to 8 PCs. Consider reformulating / being more specific.

* line 333: the usage of $\alpha=0.05$ seems unmotivated. Why 0.05? Why not 0.01 or 0.1? Is it based on some prior literature?

* Figure 2: all metric are relative, which makes them a bit hard to interpret. Would be great to see, at least in the text, how much different the methods were in terms of kNN overlap. Does the difference of 0.5 in Fig 2A correspond to 1 additional NN? to 10? Are these differences large enough to be meaningful? I think this needs to be somehow conveyed.

* Fig 2D: why is the consistency panel on the very right and not on the very left (under panel A)?

* line 359: "the results of 2C were particularly informative" -- but this paragraph does not actually draw *any* conclusion from these results. That's very confusing. If the results were so informative, then what were they? Please state in text.

* line 387: "one of our simulation..." -- which one? where in fig 2D should I look at?

* line 387: the fact that Fig 2D contains the dashed line is not mentioned in the text and is not explained in the figure caption.

* line 493: "failed to remove the confounding effect of the sequencing depth" -- not sure which result this refers to?

* line 842 and elsewhere: "delta-method-based" should I think be written with two hyphens.

* line 870: what "additional heuristics"?

- * line 878 and below: somewhat unclear what version of Pearson residuals was used for composite transformations.
- * line 907: after selecting HVGs, do you recompute everything, or simply subset the already obtained matrix?
- * line 912: not very clear, especially for Pearson residuals. Which alpha was used e.g. for the analytic Pearson residuals?
- * line 937: why do you use the global alpha here gene-specific alpha in line 912?
- * line 1004: why alpha=0.01 here? and not 0.05 as elsewhere?
- * line 1029: more details please
- * Figs S10B, S11B: do not stretch t-SNE embeddings along one axis. The aspect ratio is meaningful.
- * line 1077: in the limit $\alpha \rightarrow 0$, the formula should somehow reduce to $\sqrt{\lambda}$, which is variance-stabilizing for Poisson. Does this perhaps deserve a comment in section A1?

Reviewer #3:

Remarks to the Author:

Overall we think it is a very well written article with solid analysis and clear and relevant conclusions.

Minor point: One thing we were a bit unsure about is their reasoning for the consistency metric. It could be interesting to see a comparison to the consistency of the knn between randomly split genes of the raw data, as a kind of negative control. The simulated and down-sampled datasets show the same tendencies, so I think they have enough support for their conclusions and it is simply interesting to know.

Another minor point: Figure 1 C, x-axis label is missing

Author Rebuttal, second revision:

Point-by-point response ‘Comparison of Transformations for Single-Cell RNA-Seq Data’

Round 2, Oct 2022

Constantin Ahlmann-Eltze and Wolfgang Huber

Reviewer 1

Remarks to the Author: The manuscript has been vastly improved, it no longer reads like a review, but includes many additional analyses on relevant data-sets. For readers with specific questions, the authors provide a shiny app so that everybody can browse the benchmarking results. Finally, the discussion states clear recommendations for the bench scientist. All in all, the manuscript can be published as is.

Thank you.

Reviewer 2

Remarks to the Author: The revised version of the paper by Ahlmann-Eltze & Huber is a comprehensive benchmark of different approaches to normalization of scRNA-seq UMI data. As I wrote before, the topic is important and very timely. In the first round, all reviewers asked to make the benchmark more comprehensive, and the authors did a very good job at addressing this point. The paper, again, is very clearly written, easy to follow, and the figures are concise and masterful.

That said, I am not convinced that the presented evidence supports the authors’ conclusions (which are, as the authors admit, nontrivial and surprising). Below I describe what I think is currently missing. Without these results, the paper, in my opinion, is inconclusive. I request some simple experiments, and I think their outcome would determine whether the authors’ conclusions hold up or not.

Overall my worry is that the authors were so strict about what can be taken as “ground truth” (line 279) that they threw the baby out with the bathwater and were only left with benchmark metrics that are insufficiently sensitive. See below. But the authors may be able to convince me otherwise.

MAJOR ISSUES

- 1) The main conclusion of the paper is that log1p transform performs as well or better than the fancy alternatives. This conclusion is mainly based on Fig 2a (consistency benchmark) that shows that log1p gives more “consistent” results. However, it is logically possible that a method gives consistent but at the same time misleading results; this could e.g. be the case if log1p is strongly (but consistently) influenced by the size factor. This concern is not addressed at all.

My suggestion would be to include “negative controls” into Figure 2, namely include rows for the “raw data” (y) and maybe also the normalized data (y/s). The authors say that the UMI data need to be transformed, but we do not actually see this in Figure 2. Would y and y/s perform much worse than everything else? In particular, would they be less “consistent” in Fig 2a? If y or y/s turns out to be as consistent as the log1p, then this would, I think, suggest that the consistency metric is not very useful.

We thank the reviewer for this thoughtful and pertinent point. We included the two suggested negative controls (the raw counts y and the scaled raw counts y/s) and added them to Figure 2. The results are

consistent with the stated conclusion—namely, the performance of these (non-)transformations is among the worst. This indicates affirmatively that the consistency benchmark of Figure 2A is not dominated by a “consistent-but-wrong” scenario mentioned by the reviewer. Moreover, the benchmarks of Figs. 2B and 2C are not based on the consistency argument but on “ground-truth”, and we note that all three benchmarks broadly agree, indicating they they indeed pick up differences between the methods that are relevant for their intended applications, general, and transferable.

- 2) Related: the authors do conclude that $\log(\text{CPM}+1)$ performs “poorly” (line 488), but what is this assessment based on? In Figure 2A, $\log\text{CPM}$ performs very well, and it is not too bad in 2B and 2C either. I certainly agree that $\log\text{CPM}$ is a bad idea for UMI data, but I can hardly see it in Figure 2, and so I worry that the benchmark may not be sensitive enough to actually distinguish good normalization methods from bad ones.

We thank the reviewer for raising this point. The original sentence was indeed too unspecific. We amended the discussion to clarify that we are referring to the fact that the $\log(\text{CPM}+1)$ transformation performs consistently worse than $\log(y/s+1)$ and the acosh transformation (line 518 in *diff.pdf*).

- 3) Figure 1A shows that $\log_1 p$ transformation does not remove the confounding effect of the size factor. Why do the authors not turn it into a formal benchmark? In Figure S1 they even have a numerical metric to quantify the confounding (canonical correlation). I think it would be very interesting to see a figure in the style of Figure 2, which would show this canonical correlation for different transformations for all analyzed datasets. This is an objective performance metric, and very likely delta method would perform poorly there. This I think deserves to be shown and then discussed.

We added a panel B to Suppl. Figure S1 that shows the canonical correlations for each of the panels (methods) in Fig. S1A as a bar chart, in the same style as Figure 2. We discuss the problem that varying size factors pose for delta method-based transformations in the first paragraph of the ‘Conceptual differences’ section. The discussion of this topic is quite prominent in the manuscript.

- 4) This may be a less important issue, but I am surprised that the intersection of all kNN neighborhoods for Fig in 2C was non-empty! Line 1023: how many neighbors did you have in the intersection? On average? This needs to be reported, maybe show a histogram. Further, what fraction of cells did actually have more than one reliable nearest neighbor (line 1034)? If it was a very small fraction, does it not undermine the usefulness of this approach? Please report and discuss.

We included a new supplementary figure (S9) that shows the pairwise overlap between the methods and the number of reliable nearest neighbors for each deeply sequenced dataset. The number of reliable nearest neighbors varied by dataset between 30% of cells having more than one reliable nearest neighbor in the HEK dataset to 99% in the mcSCRB dataset.

We also included a variation of the downsampling benchmark (Suppl. Fig. S10), where we only consider the two best performing transformations per approach. Here, we only take the intersection of eight transformations and thus find a larger number reliable nearest neighbors. The original and the reduced downsampling benchmarks show the same trends, a results that counterargues the hypothesis that the downsampling benchmark is undermined by small numbers of reliable nearest neighbors.

MINOR ISSUES

- Line 233: valid argument, but the first and the third histograms in Figure 1C are very obviously *not* a linear transformation of each other. Why? I assume it is because Pearson residuals are linear transformation only assuming that all size factors are equal to 1, while in Fig 1C they are likely not. I think this needs to be clarified in the text, and maybe also illustrated somehow (additional row in Figure 1 that would be similar to 1C but only use cells with nearly the same size factors? or maybe insert the figure from the rebuttal into the supplementary materials?)

Yes, the reviewer is correct that the Pearson residuals are linear transformations of the size factor-scaled

counts and not of the raw counts. We now include the panel for the size factor-scaled counts in Suppl. Fig. S3 which show the same shape as the Pearson residuals and adjusted the text to point this out (line 243 in diff.pdf).

- line 335: “four to eight dimension settings for the PCA” – this is unclear. This reads as if you used up to 8 PCs. Consider reformulating / being more specific.

Thank you. We reformulated this (line 344 in diff.pdf).

- line 333: the usage of $\alpha=0.05$ seems unmotivated. Why 0.05? Why not 0.01 or 0.1? Is it based on some prior literature?

We chose $\alpha=0.05$ based on the range of estimates we saw in past practice; however, $\alpha=0.01$ or $\alpha=0.1$ would also have been reasonable values. Our benchmarks show (see Shiny app) that the specific choice of this number (within a wide range) does not much alter the results.

- Figure 2: all metrics are relative, which makes them a bit hard to interpret. Would be great to see, at least in the text, how much different the methods were in terms of kNN overlap. Does the difference of 0.5 in Fig 2A correspond to 1 additional NN? to 10? Are these differences large enough to be meaningful? I think this needs to be somehow conveyed.

All results underlying the benchmark in Figure 2 are provided in Suppl. Fig. S8. on an absolute scale. We added a paragraph (line 384 in diff.pdf) that discusses the relevance of the observed kNN overlap differences.

- Fig 2D: why is the consistency panel on the very right and not on the very left (under panel A)?

Fixed.

- line 359: “the results of 2C were particularly informative” – but this paragraph does not actually draw *any* conclusion from these results. That’s very confusing. If the results were so informative, then what were they? Please state in text.

Fixed.

- line 387: “one of our simulation...” – which one? where in fig 2D should I look at?

We have rephrased the sentence to reflect that we here refer to the Random Walk simulation (line 417 in diff.pdf).

- line 387: the fact that Fig 2D contains the dashed line is not mentioned in the text and is not explained in the figure caption.

The dashed line shows the results of Sanity Distance and this is explained in the figure caption. We visually distinguish the results for Sanity Distance because they do not depend on the number of PCA dimensions.

- line 493: “failed to remove the confounding effect of the sequencing depth” – not sure which result this refers to?

We were referring to the results in Suppl. Fig. S1 and the fact that the first 10 PCs of the scaled shifted logarithm had a canonical correlation of 0.71 with the size factor. We have added a reference to the paragraph to make this clearer (line 523 in diff.pdf).

- line 842 and elsewhere: “delta-method-based” should I think be written with two hyphens.

There was some discussion of a similar case on Stackexchange (<https://english.stackexchange.com/a/91206>). Based on this, we decided to go with the single hyphen (“delta method-based”) spelling, but will also be happy to take advice from the journal editors.

- line 870: what “additional heuristics”?

We were referring to fact that `sctransform V2`, in addition to applying Eq. (4), for example, subsamples the cells, excludes genes with a variance smaller than the expected Poisson variance, and fixes the maximum variance for a non-zero median UMI to 5.

- line 878 and below: somewhat unclear what version of Pearson residuals was used for composite transformations.

We used the `transformGamPoi`-based Pearson residuals for the composite transformations. We updated the paragraph to reflect that (line 916 in `diff.pdf`).

- line 907: after selecting HVGs, do you recompute everything, or simply subset the already obtained matrix?

The HVG selection step is applied to the transformed counts. The transformation is not recomputed after the HVG selection. We amended the paragraph in the Methods section to clarify this (line 885 in `diff.pdf`).

- line 912: not very clear, especially for Pearson residuals. Which alpha was used e.g. for the analytic Pearson residuals?

We used a gene-specific overdispersion estimate for all residual-based transformations. We rephrased the paragraph to clarify this choice.

- line 937: why do you use the global alpha here gene-specific alpha in line 912?

We thank the reviewer for spotting the inconsistency. We are now using gene specific overdispersion estimates for all applicable transformation for both Suppl. Fig. S1 and S2.

- line 1004: why $\alpha=0.01$ here? and not 0.05 as elsewhere?

We used a different overdispersion setting here, to deviate from the value used in some of the transformation methods, and thus to make it a little harder for them and to not appear “favoring” them.

- line 1029: more details please

Fixed.

- Figs S10B, S11B: do not stretch t-SNE embeddings along one axis. The aspect ratio is meaningful.

Fixed. We thank the reviewer for catching us with one of our own pet pies!

- line 1077: in the limit $\alpha \rightarrow 0$, the formula should somehow reduce to $\sqrt{\cdot}$, which is variance-stabilizing for Poisson. Does this perhaps deserve a comment in section A1?

We thank the reviewer for this suggestion and have added a sentence to explain this relation between the `acosh` transformation and the square root transformation (line 1132 in `diff.pdf`).

Reviewer 3:

Remarks to the Author: Overall we think it is a very well written article with solid analysis and clear and relevant conclusions.

Minor point: One thing we were a bit unsure about is their reasoning for the consistency metric. It could be interesting to see a comparison to the consistency of the knn between randomly split genes of the raw data, as a kind of negative control. The simulated and down-sampled datasets show the same tendencies, so I think they have enough support for their conclusions and it is simply interesting to know.

We included the suggested negative control in all three benchmarks. In line with our expectation, the consistency of raw counts (and the raw counts scaled by the size factor) is lower than for the other transformations.

Another minor point: Figure 1 C, x-axis label is missing

Fixed.

Decision Letter, third revision:

11th Nov 2022

Dear Dr. Huber,

Thank you for submitting your revised manuscript "Comparison of Transformations for Single-Cell RNA-Seq Data" (NMETH-AS46949C). It has now been seen by the original referee and their comments are below. The reviewers find that the paper has improved in revision, and therefore we'll be happy in principle to publish it in Nature Methods, pending minor revisions to satisfy the referees' final requests and to comply with our editorial and formatting guidelines.

TRANSPARENT PEER REVIEW

ORCID

Thank you again for your interest in Nature Methods. Please do not hesitate to contact us if you have any questions. We will be in touch again soon.

[REDACTED]

Best regards,
Lei
Lei Tang, Ph.D.

Senior Editor
Nature Methods

Reviewer #2 (Remarks to the Author):

I appreciate the authors' responses, and think the manuscript can be published almost as is. I list three minor phrasing suggestions.

1) I really like the new Fig S1B (but note that this panel is not described in the Fig S1 caption). I personally would prefer if it were a main figure in the main text. But even if it stays supplementary, I think it could be mentioned in the main text explicitly (currently it isn't), for example in Results and/or Discussion, e.g. in line 351 one could add "despite being more affected by the size factor confounding (Fig S1B)". The same remark could also be added to line 500. Currently it looks a bit like brushing this inconsistency a little under the carpet.

2) line 362: "agreed well with the trends observed in the simulation and the consistency" -- to be honest, panel A shows good performance of log while panel B shows that everything is the same. IMHO this sentence is unclear and not precise. Why not write explicitly, that panel C shows equally good performance of log and of Pearson residuals? This is an important conclusion but is currently not spelled out.

3) line 485: "shifted logarithm performed as well for cells with particularly large or small size factors": I see increasing trend (and not a constant flat line) in panel D left and panel D right. Perhaps I am misunderstanding this sentence, so may be worth reformulating.

Author Rebuttal, third revision:

Point-by-point response ‘Comparison of Transformations for Single-Cell RNA-Seq Data’

Round 3, Jan 2023

Constantin Ahlmann-Eltze and Wolfgang Huber

Reviewer #2 (Remarks to the Author):

I appreciate the authors’ responses, and think the manuscript can be published almost as is. I list three minor phrasing suggestions.

- 1) I really like the new Fig S1B (but note that this panel is not described in the Fig S1 caption). I personally would prefer if it were a main figure in the main text. But even if it stays supplementary, I think it could be mentioned in the main text explicitly (currently it isn’t), for example in Results and/or Discussion, e.g. in line 351 one could add “despite being more affected by the size factor confounding (Fig S1B)”. The same remark could also be added to line 500.

We thank the reviewer for their input and for spotting our oversight not to include a description of Panel B in the caption of Suppl. Fig. S1. We amended the caption and also included an explicit reference to Suppl. Fig. S1B in the first paragraph of the section “Conceptual differences” and in the discussion (lines 196 and 526 in diff.pdf).

Currently it looks a bit like brushing this inconsistency a little under the carpet.

We fully agree with the reviewer on the importance of the point that size factor variability affects the different transformation approaches differently, and delta method-based transformations particularly badly. We do not think that we sweep it under the carpet. Indeed, we highlight it at several places:

- First panel of Fig. 1
- First paragraph of the ‘Conceptual differences’ section
- We highlight that the other transformation approaches are less affected (line 197 and line 525 in diff.pdf).

The main focus of our manuscript is the empirical performance of the different transformations, based on the benchmark criteria outlined in Section “Benchmarks”. Turning Panel B of Suppl. Fig. S1 into a main figure would however elevate its topic (influence of size factor on PCA embedding) effectively to another benchmark criterion, and we are concerned that this would dilute the other, in our view more substantial and consequential, benchmarks. Therefore, we would prefer to balance the presentation as it is.

- 2) line 362: “agreed well with the trends observed in the simulation and the consistency” – to be honest, panel A shows good performance of log while panel B shows that everything is the same. IMHO this sentence is unclear and not precise.

The reviewer is correct in pointing out that the absolute differences in Panel B (simulation) are less pronounced than those in Panels A (consistency) and C (downsampling). Nonetheless, the rankings still agree: e.g., one can quantify this by considering the correlations between the metrics from the three benchmarks. The Pearson correlations of B with A and C are 0.75 and 0.81, respectively, the Spearman correlations are 0.79 and 0.57. These correlations are all “significantly” different from 0 with $p < 0.005$. We would therefore contend that the sentence is correct and that additional qualifications, while always possible, would lengthen the text, with limited benefit.

Why not write explicitly, that panel C shows equally good performance of log and of Pearson residuals? This is an important conclusion but is currently not spelled out.

Indeed, we do write this point explicitly in the discussion in line 521: “The Pearson residuals-based transformation has attractive theoretical properties and, in our benchmarks, performed similarly well as the shifted logarithm transformation”.

We compare the performance of many transformations in Figure 4A, titled “ $\log(y/s+1)$ performs on par or better than alternative transformations”, including an explicit comparison of the log transformation and the Pearson residuals (right-most panel in the first row). As the reviewer notes, the panel shows equivalent performance of the log transformation and the Pearson residuals for the downsampling benchmark. Altogether, we think that we present this result rather prominently, in the text as well as in multiple figure panels.

3) line 485: “shifted logarithm performed as well for cells with particularly large or small size factors”: I see increasing trend (and not a constant flat line) in panel D left and panel D right. Perhaps I am misunderstanding this sentence, so may be worth reformulating.

We thank the reviewer for spotting this ambiguous wording. We meant that the log transformation did not perform worse than other transformations for particularly large or small cells. We have modified the sentence to reflect this (line 485 in diff.pdf). It now says: “yet on three datasets, the shifted logarithm did not perform worse than other transformations for cells with particularly large or small size factors (Fig. 4D).”